# MARCH5 mediates NOXA-dependent MCL1 degradation driven by kinase inhibitors and integrated stress response activation

Seiji Arai[1,2†], Andreas Varkaris[1†], Mannan Nouri[1], Sen Chen[1], Lisha Xie[1], Steven P Balk[1]*

[1]Hematology-Oncology Division, Department of Medicine, and Cancer Center, Beth Israel Deaconess Medical Center and Harvard Medical School, Boston, United States; [2]Department of Urology, Gunma University Hospital, Maebashi, Japan

**Abstract** MCL1 has critical antiapoptotic functions and its levels are tightly regulated by ubiquitylation and degradation, but mechanisms that drive this degradation, particularly in solid tumors, remain to be established. We show here in prostate cancer cells that increased NOXA, mediated by kinase inhibitor activation of an integrated stress response, drives the degradation of MCL1, and identify the mitochondria-associated ubiquitin ligase MARCH5 as the primary mediator of this NOXA-dependent MCL1 degradation. Therapies that enhance MARCH5-mediated MCL1 degradation markedly enhance apoptosis in response to a BH3 mimetic agent targeting BCLXL, which may provide for a broadly effective therapy in solid tumors. Conversely, increased MCL1 in response to MARCH5 loss does not strongly sensitize to BH3 mimetic drugs targeting MCL1, but instead also sensitizes to BCLXL inhibition, revealing a codependence between MARCH5 and MCL1 that may also be exploited in tumors with *MARCH5* genomic loss.

**\*For correspondence:**
sbalk@bidmc.harvard.edu

[†]These authors contributed equally to this work

**Competing interests:** The authors declare that no competing interests exist.

## Introduction

Androgen deprivation therapy to suppress activity of the androgen receptor (AR) is the standard treatment for metastatic prostate cancer (PCa), but tumors invariably recur (castration-resistant prostate cancer, CRPC). The majority will initially respond to agents that further suppress AR, but most men relapse within 1–2 years and these relapses appear to be driven by multiple AR dependent and independent mechanisms (*Watson et al., 2015*; *Yuan et al., 2014*), which may include increased expression of anti-apoptotic proteins. The anti-apoptotic BCL2 family proteins (including BCL2, BCLXL, and MCL1) act by neutralizing BAX and BAK, and by inhibiting the BH3-only pro-apoptotic proteins that can activate BAX/BAK (primarily BIM) (*Montero and Letai, 2018*). These interactions are mediated by the BH3 domain, and BH3-mimetic drugs can enhance apoptosis by mimicking the activity of BH3-only pro-apoptotic proteins and thereby antagonizing the anti-apoptotic BCL2 family proteins (*Merino et al., 2018*; *Knight et al., 2019*). ABT-737 (*Oltersdorf et al., 2005*) and ABT-263 (navitoclax, orally bioavailable analogue of ABT-737) (*Tse et al., 2008*) are BH3-mimetics that directly bind to BCL2, BCLXL, and BCLW (but not MCL1), which blocks their binding to pro-apoptotic BH3 only proteins such as BIM and their ability to neutralize BAX/BAK. Navitoclax has single-agent activity in hematological malignancies (*Roberts et al., 2012*), but causes thrombocytopenia due to BCLXL inhibition. A BCL2-specific agent that spares platelets (ABT-199, venetoclax) is similarly active and is now FDA approved for chronic lymphocytic leukemia (*Pan et al., 2014*; *Roberts et al., 2016*).

In contrast, most solid tumors are resistant to these agents (*Faber et al., 2015*), which appears to reflect an important role for MCL1 (*Faber et al., 2015*; *van Delft et al., 2006*; *Konopleva et al., 2006*; *Santer et al., 2015*; *Williams et al., 2017*; *Xiao et al., 2015*). Indeed, preclinical studies indicate that navitoclax may be efficacious in solid tumors when used in combination with other agents acting through a variety of mechanisms, including by decreasing MCL1 expression (*Faber et al., 2015*; *Xiao et al., 2015*; *Leverson et al., 2015*; *Chen et al., 2011*; *Modugno et al., 2015*; *Anderson et al., 2016*; *Tong et al., 2017*; *Arai et al., 2018*). BH3 mimetics that target MCL1 (including AMG176, S63845 and AZD5991) are now becoming available and may have single agent activity in a subset of tumors (*Kotschy et al., 2016*; *Ashkenazi et al., 2017*; *Letai, 2016*; *Merino et al., 2017*; *Tron et al., 2018*; *Caenepeel et al., 2018*), but efficacy in most solid tumors will likely still require combination therapies (*Merino et al., 2018*; *Kotschy et al., 2016*; *Merino et al., 2017*). Moreover, the toxicities associated with direct MCL1 antagonists, alone or in combination therapies, remain to be determined.

We reported previously that navitoclax (acting through BCLXL blockade), in combination with several kinase inhibitors (erlotinib, lapatinib, cabozantinib, sorafenib) could induce rapid and marked apoptotic responses in PCa cells (*Arai et al., 2018*). This response was preceded by a dramatic increase in MCL1 degradation, and we confirmed that navitoclax could drive apoptotic responses in vitro and in vivo in PCa cell that were depleted of MCL1 by RNAi or CRISPR. Significantly, the enhanced MCL1 degradation in response to kinase inhibitors was not mediated by well-established mechanisms including through GSK3β-mediated phosphorylation (and the downstream ubiquitin ligases βTrCP or Fbw7), or by the ubiquitin ligase HUWE1/MULE that has been reported to mediate both basal MCL1 degradation and MCL1 degradation in response to DNA damage and NOXA binding (*Gomez-Bougie et al., 2011*; *Guikema et al., 2017*; *Zhong et al., 2005*; *Warr et al., 2005*).

In this study, we found that treatment with kinase inhibitors initiates an integrated stress response (ISR) leading to increased ATF4 protein and subsequent increased transcription of NOXA, and that the enhanced degradation of MCL1 was NOXA-dependent. We further identified the mitochondria-associated ubiquitin ligase MARCH5 as the mediator of this stress-induced and NOXA-dependent MCL1 degradation. MARCH5 is a RING-finger E3 ligase with an established function in mediating the ubiquitylation and degradation of several proteins that regulate mitochondrial fission and fusion (*Chen et al., 2017*; *Park et al., 2014*; *Yonashiro et al., 2006*; *Xu et al., 2016*; *Cherok et al., 2017*). MARCH5 depletion both abrogated the decrease in MCL1 in response to cellular stress and substantially increased basal MCL1 in multiple epithelial cancer cell lines, indicating that MARCH5 makes a major contribution to regulating MCL1 levels under basal conditions and in responses to cellular stress. Significantly, while the MARCH5 mediated degradation of MCL1 markedly sensitized tumor cells to BCLXL inhibition, MARCH5 depletion, which occurs in ~5% of PCa, also sensitized to BCLXL inhibition despite increased MCL1, revealing a codependency between MCL1 and MARCH5. Together these results reveal therapeutic opportunities for the use of agents targeting BCLXL in solid tumors.

## Results

### NOXA upregulation mediates increased MCL1 degradation in PCa cells

As we reported previously, multiple kinase inhibitors including the EGFR inhibitor erlotinib could rapidly (within 4 hr) and markedly enhance the proteasome-dependent degradation of MCL1 (*Figure 1—figure supplement 1A,B*). Moreover, we found that this occurred by a mechanism that was independent of the ubiquitin ligase HUWE1 (MULE) and of ubiquitin ligases downstream of GSK3β (βTRCP, FBW7) (*Arai et al., 2018*). BIM and NOXA are the primary BH3-only proteins that bind MCL1, and can increase or decrease its stability, respectively (*Warr et al., 2005*; *Willis et al., 2005*; *Czabotar et al., 2007*). Consistent with our previous results, 4 hr treatment with erlotinib did not decrease BIM, indicating that loss of BIM is not a basis for the marked decrease in MCL1 protein (*Figure 1A*). In contrast, NOXA expression was increased by erlotinib, suggesting this may drive the increased MCL1 degradation. Indeed, depleting NOXA with three different siRNA suppressed this decrease in MCL1 (*Figure 1B*). Moreover, more complete depletion of NOXA with the pooled siRNAs prevented the erlotinib-mediated MCL1 reduction, indicating a NOXA-dependent mechanism for decreasing MCL1 (*Figure 1C*). In contrast, while depletion of BIM by siRNA caused a

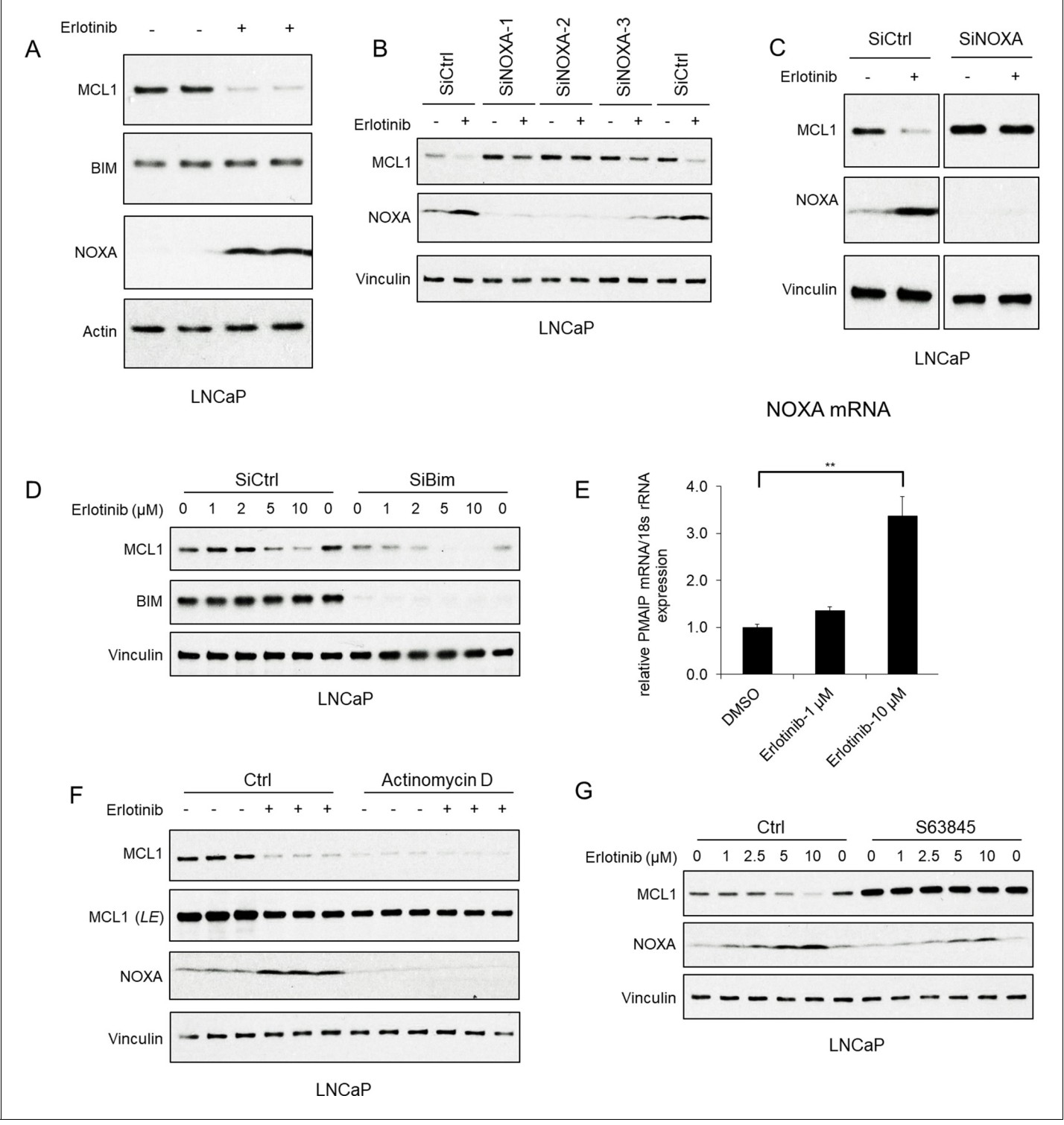

**Figure 1.** EGFR inhibition decreases MCL1 via NOXA-dependent mechanism. (**A**) LNCaP cells were treated with EGFR inhibitor erlotinib (10 μM) for 3 hr, followed by immunoblotting. (**B**) LNCaP cells were transfected with three distinct NOXA siRNA or non-target control siRNA for 3 days, then were treated with erlotinib (10 μM) for 3 hr. (**C**) LNCaP cells were transfected with pooled NOXA siRNAs or non-target control siRNA for 3 days, then were treated with erlotinib (10 μM) for 5 hr. (**D**) LNCaP cells transfected with pooled BIM siRNAs or non-target control siRNA were treated with erlotinib (0–10 μM) for 3 hr. (**E**) LNCaP cells were treated with erlotinib (0–10 μM) for 2 hr, followed by NOXA (*PMAIP*) mRNA measurement by qRT-PCR. Data reflect biological triplicates with each mRNA sample assayed in duplicate (technical replicate). 18 s rRNA was used as an internal control. (\*\*, p<0.01). (**F**) LNCaP cells were pretreated with RNA synthesis inhibitor actinomycin D (10 μg/ml) for 30 min, followed by treatment with erlotinib (10 μM) for 3 hr. LE,

*Figure 1 continued on next page*

*Figure 1 continued*

long exposure. (G) LNCaP cells were pretreated with MCL1 inhibitor S63845 (500 nM) for 3 hr, followed by treatment with erlotinib (0–10 μM) for 3 hr. Immunoblots are representative of results obtained in at least three independent experiments.

The online version of this article includes the following figure supplement(s) for figure 1:

**Figure supplement 1.** EGFR inhibition increases proteasome-dependent MCL1 degradation.

decrease in basal MCL1, it did not prevent the further decrease in MCL1 in response to erlotinib (*Figure 1D*).

Erlotinib rapidly (within 2 hr) upregulated NOXA mRNA (*Figure 1E*), indicating a transcriptional mechanism for increasing NOXA protein. Consistent with this finding, inhibiting new synthesis of mRNA with actinomycin D decreased basal NOXA protein, and prevented the erlotinib-mediated upregulation of NOXA (*Figure 1F*). Actinomycin D similarly decreased basal MCL1 protein expression through transcriptional repression, but importantly prevented the erlotinib-mediated MCL1 reduction (*Figure 1F*).

BH3-mimetic agents that occlude the BH3 binding site of MCL1, and would therefore prevent binding of BIM and NOXA, have recently been developed (*Kotschy et al., 2016*; *Tron et al., 2018*; *Caenepeel et al., 2018*). Therefore, we tested whether one such agent (S63845), by competing with NOXA for binding to MCL1, could prevent the erlotinib-mediated decrease in MCL1. Significantly, S63845 increased basal MCL1 expression and prevented the erlotinib-mediated decrease in MCL1 (*Figure 1G*). Together, these data show that erlotinib induces transcriptional upregulation of NOXA, and indicate that this increase in NOXA is directly enhancing MCL1 degradation.

## NOXA upregulation is mediated by the integrated stress response

To determine how erlotinib was increasing NOXA transcription we first focused on p53, as NOXA is a major transcriptional target of p53. However, treatment with erlotinib did not cause any change in p53 expression (*Figure 2A*; *Figure 1—figure supplement 1A*), indicating a p53-independent mechanism for increasing NOXA mRNA. The alternative p53-independent pathway that may increase NOXA transcription is the integrated stress response (ISR), which can be triggered by factors including hypoxia, glucose or amino acid depletion, genotoxic stress, and the endoplasmic reticulum stress/unfolded protein response (*Pakos-Zebrucka et al., 2016*). These stresses activate kinases including PERK (in response to endoplasmic reticulum stress), GCN2 (in response to amino acid starvation), and PKR (in response dsRNA and additional cellular stresses), which converge on phosphorylation of eIF2α (*Guikema et al., 2017*; *Armstrong et al., 2010*; *Albershardt et al., 2011*; *Wang et al., 2009*). Consistent with ISR activation, we found that erlotinib rapidly (within 30 min) increased phosphorylation of eIF2α (*Figure 2B*). Both PERK and GCN2 appear to be contributing to this ISR activation as MCL1 degradation in response to erlotinib was prevented by siRNA targeting PERK and GCN2 in combination, but not by either alone (*Figure 2—figure supplement 1*).

The phosphorylation of eIF2α causes an increase in translation of the transcription factor ATF4, which can then stimulate the expression of multiple genes to either resolve the cellular stress or drive to apoptosis. Indeed, eIF2α phosphorylation in response to erlotinib was associated with an increase in ATF4 protein (*Figure 2B*). With respect to NOXA, ATF4 has been reported to stimulate expression of NOXA, either directly or as a heterodimer with ATF3 (41,43), although this is generally observed after prolonged stress. Nonetheless, an increase in NOXA protein was observed after 60–90 min of erlotinib treatment, and this rapid time course coincided with the increase in ATF4 and decrease in MCL1 (*Figure 2B*). Moreover, treatment with an ISR inhibitor (ISRIB), which suppresses the effects of eIF2α phosphorylation (*Sidrauski et al., 2013*), decreased basal ATF4 and suppressed the erlotinib-mediated increase in ATF4 and NOXA, providing further evidence for this pathway (*Figure 2C*). Consistent with these findings, ISRIB suppressed the erlotinib-mediated increase in NOXA mRNA (*Figure 2D*), while MCL1 mRNA was unaffected by these treatments (*Figure 2E*). Together, these findings indicate that activation of the ISR by erlotinib drives the rapid induction of NOXA, which then promotes MCL1 degradation.

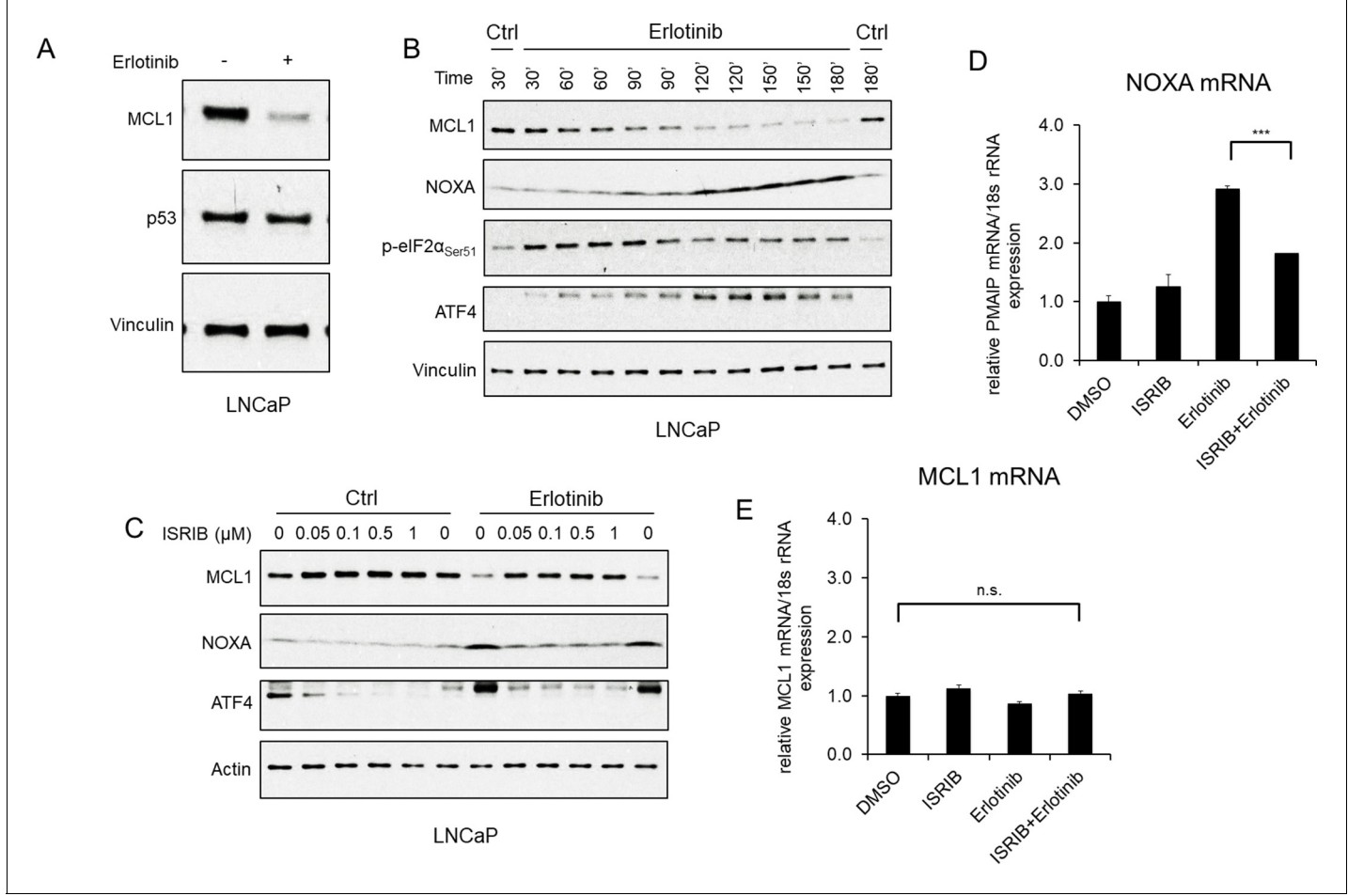

**Figure 2.** EGFR inhibition upregulates NOXA through ISR activation. (**A**) LNCaP cells were treated with erlotinib (10 µM) for 3 hr, followed by immunoblotting. (**B**) LNCaP cells were treated with erlotinib (10 µM) at time 0 and were harvested over a time course from 30 to 180 min. (**C**) LNCaP cells were treated with ISR inhibitor ISRIB trans-isomer (0–1 µM) for 1 hr, followed by treatment with erlotinib (10 µM) for 3 hr. The weak band migrating just above the major ATF4 band was proportional to the major band and may reflect a posttranslational modification. (**D and E**) LNCaP cells were pretreated with ISRIB trans-isomer (100 nM) or DMSO for 1 hr, followed by erlotinib (10 µM) or DMSO for 2 hr. NOXA (*PMAIP*) mRNA (**D**) and *MCL1* mRNA (**E**) were measured by qRT-PCR. Data reflect biological triplicates with each mRNA sample assayed in duplicate (technical replicate). 18 s rRNA was used as an internal control. (n.s., not significant; ***, p<0.001). Immunoblots in (**A**) and (**C**) are representative of results obtained in three independent experiments, and (**B**) is representative of two independent experiments.

The online version of this article includes the following figure supplement(s) for figure 2:

**Figure supplement 1.** LNCaP cells were transfected with siRNA pools targeting PERK, GCN2, the combined PERK and GCN2 pools, or a nontarget control siRNA (siNC).

## MARCH5 mediates kinase inhibitor/NOXA-dependent MCL1 degradation

We next sought to identify E3 ligases that contribute to kinase inhibitor-mediated and NOXA-dependent MCL1 degradation. MCL1 is a substrate for the ubiquitin ligase HUWE1 (MULE), and HUWE1 has been reported to mediate MCL1 degradation by NOXA (*Gomez-Bougie et al., 2011*; *Guikema et al., 2017*; *Zhong et al., 2005*). However, we reported previously that while HUWE1 depletion could increase basal MCL1 levels, it did not prevent the increased degradation of MCL1 in response to kinase inhibitors (*Arai et al., 2018*). *Figure 3—figure supplement 1A* shows that HUWE1 depletion does not affect the erlotinib-mediated increase in NOXA, and that it does not prevent the subsequent decrease in MCL1.

NEDD8 conjugation is essential for cullin-dependent E3 ligases to ubiquitylate their substrates. To determine the role of cullin-dependent E3 ligases in MCL1 degradation in response to tyrosine

kinase inhibition, we examined whether NEDD8 inhibition could prevent the effect of tyrosine kinase inhibitors. Treatment with NEDD8 inhibitor MLN4924 increased p27 (a known target of cullin-dependent E3 ligase CUL4), but did not increase MCL1 or block the effects of erlotinib (*Figure 3—figure supplement 1B*). Indeed, MLN4924 moderately decreased MCL1 protein, which may be due to an increase in NOXA, whose degradation is mediated by a cullin-dependent E3 ligase (*Zhou et al., 2017*). MLN4924 similarly failed to prevent the decrease in MCL1 in response to lapatinib (EGFR/ERBB2 inhibitor) (*Figure 3—figure supplement 1C*), indicating that a cullin-independent mechanism is driving the increased MCL1 degradation.

We then hypothesized that a cullin-independent E3 ligase that localizes to mitochondria, where MCL1 is mainly located, may promote MCL1 degradation in response to tyrosine kinase inhibition. To assess this hypothesis, we first examined the well-known mitochondria-associated cullin-independent E3 ligase PARKIN, which has been implicated as a ubiquitin ligase for MCL1 (*Carroll et al., 2014*). However, while PARKIN depletion increased its target protein p62, it did not increase MCL1 or block the effect of erlotinib (*Figure 3—figure supplement 1D*). MARCH5 is another mitochondria-associated cullin-independent E3 ligase that has been implicated as a regulator of MCL1 (37,47). Significantly, depleting MARCH5 with siRNA increased basal expression of MCL1 and a known MARCH5 substrate, MiD49, in LNCaP cells (*Figure 3A*; *Figure 3—figure supplement 1E*). MARCH5 depletion did not increase MCL1 mRNA (*Figure 3—figure supplement 1F*), further supporting a posttranscriptional mechanism for increasing MCL1. MARCH5 depletion also increased basal MCL1 in PC3 PCa cells (*Figure 3B*) and in additional prostate, breast and lung cancer cell lines (*Figure 3—figure supplement 1G–L*). These results show that MARCH5 is a major mediator of basal MCL1 degradation in epithelial cancer cell lines.

Significantly, MARCH5 depletion prevented the decrease in MCL1 by erlotinib and cabozantinib (C-MET/VEGFR2 inhibitor) in LNCaP cells (*Figure 3A,B*), and in other prostate cancer cells (PC3 and DU145 cells) (*Figure 3C*, *Figure 3—figure supplement 1M*), indicating that the decreases in MCL1 by these kinase inhibitors are mediated by MARCH5. In contrast, MARCH5 depletion did not prevent MCL1 loss in cells treated with dinaciclib (*Figure 3A*), which decreases MCL1 mRNA through inhibition of CDK9 and subsequent decrease in MCL1 transcription. To confirm these findings, we then used CRISPR/CAS9 to delete MARCH5. Consistent with the RNAi results, there was a marked increase of MCL1 expression, as well of the MARCH5 substrate MiD49, in each of three MARCH5 depleted lines (*Figure 3D*). Moreover, erlotinib no longer decreased MCL1 in these MARCH5 depleted lines (*Figure 3D*). As expected, transient overexpression of exogenous MARCH5 decreased MCL1 in control and MARCH5 depleted cells (*Figure 3E*).

Interestingly, and consistent with a previous report (*Subramanian et al., 2016*), MARCH5 depletion by CRISPR or siRNA also increased NOXA protein in LNCaP cells (*Figure 3D*; *Figure 4—figure supplement 1A*, respectively), and in additional cell lines (*Figure 3—figure supplement 1G–L*). MARCH5 depletion did not increase, but instead decreased NOXA mRNA (*Figure 4—figure supplement 1B*), indicating this increase in NOXA protein is through a post-transcriptional mechanism. One plausible mechanism is through increased binding to MCL1, as a previous study found that MCL1 could protect NOXA from proteasome-mediated degradation (*Craxton et al., 2012*). Consistent with this mechanism, the increased levels of NOXA and of BIM in MARCH5 depleted cells coincided with increased binding of these proteins to MCL1 (*Figure 3F*). To further assess this mechanism, we treated cells with an MCL1-targeted BH3 mimetic agent, S63845, to interfere with BH3 domain-mediated interactions with MCL1. Significantly, S63845 decreased both NOXA and BIM in the MARCH5-depleted cells, consistent with them being stabilized by MCL1 (*Figure 3G*). Of note, S63845 increased MCL1 in both the control and MARCH5 depleted cells, indicating that additional ubiquitin ligases may partially compensate for MARCH5 loss in driving basal MCL1 degradation.

We also examined the effects on NOXA and BIM of depleting or overexpressing MCL1. Cells with CRISPR-mediated MCL1 depletion had markedly reduced NOXA and BIM, providing further evidence that MCL1 protects both from degradation (*Figure 3H*). Conversely, NOXA and BIM were increased in cells that overexpress ectopic MCL1 (*Figure 3I*). However, while MCL1 levels were comparable in cells overexpressing ectopic MCL1 and in MARCH5-depleted cells, the increases in NOXA and BIM were greater in the latter MARCH5-depleted cells. One explanation for this difference with respect to NOXA (and possibly BIM) is that the MARCH5-mediated ubiquitylation of MCL1 in MCL1-NOXA complexes may drive degradation of the complex, or be coupled to the degradation of NOXA mediated by a distinct ubiquitin ligase.

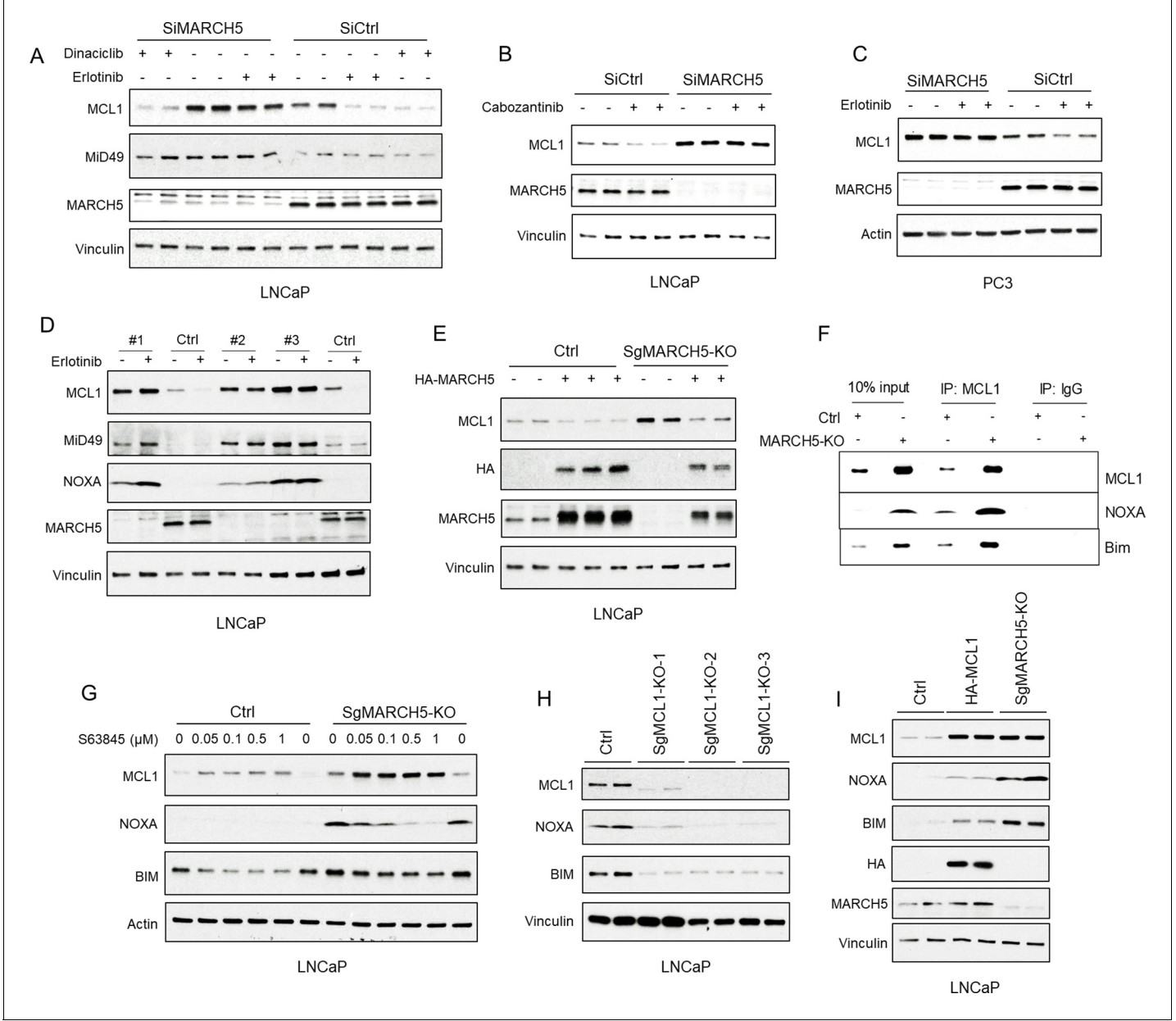

**Figure 3.** Tyrosine kinase inhibitors decrease MCL1 via mitochondria-associated E3 ligase MARCH5. (**A**) LNCaP cells transfected with pooled MARCH5 siRNAs or non-target control siRNA were treated with DMSO, erlotinib (10 μM), or dinaciclib (200 nM) for 4 hr. (**B**) LNCaP cells transfected with pooled MARCH5 siRNAs or non-target control siRNA were treated with multi-kinase inhibitor cabozantinib (5 μM) for 5 hr. (**C**) PC3 cells transfected with pooled MARCH5 siRNAs or non-target control siRNA were treated with erlotinib (10 μM) for 4 hr. (**D**) Three MARCH5-deficient LNCaP subclones generated with CRISPR/CAS9 and guide RNAs (SgMARCH5-KO #1–3), and one negative control clone (Ctrl), were treated with erlotinib (10 μM) for 4 hr. (**E**) SgMARCH5-KO or control LNCaP cells were transiently transfected with HA-tagged MARCH5, followed by immunoblotting. (**F**) Cell lysates of SgMARCH5-KO or control LNCaP with same protein amounts were subject to immunoprecipitation using anti-MCL1 rabbit antibody or control rabbit IgG with protein A agarose, followed by immunoblotting with mouse antibodies targeting for indicated proteins. (**G**) SgMARCH5-KO or control LNCaP cells were treated with S63845 (0–1 μM) for 12 hr. (**H**) Three MCL1-deficient LNCaP subclones generated with CRISPR/CAS9 and guide RNAs (MCL1-KO-1–3) or one negative control clone (Ctrl) were lysed and were immunoblotted for indicated proteins. (**I**) SgMARCH5-KO, HA-tagged MCL1 overexpressing (HA-MCL1), or control LNCaP cells were lysed and immunoblotted for indicated proteins. Immunoblot in B is representative of results obtained in two independent experiments, and the remainder are representative of at least three independent experiments.

The online version of this article includes the following figure supplement(s) for figure 3:

**Figure supplement 1.** MARCH5 knockdown increases MCL1 in additional PCa, breast, and lung cancer cell lines.

## EGFR inhibition does not alter MARCH5 expression or activity

The above findings indicated that increased NOXA in response to erlotinib was driving the MARCH5-mediated ubiquitylation and degradation of MCL1. Consistent with this conclusion, we found by coimmunoprecipitation that erlotinib treatment, in combination with proteasome inhibition, enhanced the interaction between MARCH5 and MCL1 (*Figure 4A*). Importantly, phosphorylation of BIM and NOXA can modulate their interaction with MCL1, suggesting that kinase inhibitors may further be enhancing MCL1 ubiquitylation and degradation through effects on phosphorylation of BIM, NOXA, or MCL1 that modulate NOXA/BIM-MCL1 interactions (*Tong et al., 2018*; *Ewings et al., 2007*; *Conage-Pough and Boise, 2018*). We have previously found that erlotinib did not alter MCL1 phosphorylation at sites that have been shown to enhance its ubiquitylation and degradation (*Arai et al., 2018*). To further assess the role of phosphorylation in erlotinib-mediated MCL1 degradation, we used phospho-tag gels and examined the phosphorylation state of these proteins. Erlotinib treatment did not have any clear effects on the phosphorylation of MCL1, BIM, or NOXA in LNCaP cells cultured under standard conditions in medium with fetal bovine serum (FBS) (*Figure 4B*). As a positive control for detection of phosphorylation we growth arrested LNCaP cells by culturing in medium with charcoal stripped serum (CSS), which is depleted of steroids and growth factors. As expected, EGF stimulation of these cells caused a dramatic increase in BIM phosphorylation (*Figure 4B*).

In parallel with the above experiments, we asked directly whether erlotinib enhances MCL1 interaction with NOXA versus BIM. This was assessed in MARCH5 knockout cells to avoid effects due to increased MCL1 interaction with MARCH5 by erlotinib. Erlotinib treatment did not clearly enhance MCL1 binding of NOXA versus BIM (*Figure 4C*). As expected, treatment with S63845 decreased both NOXA and BIM binding to MCL1.

We next asked whether there were alterations in MARCH5 expression or activity that may be enhancing its ubiquitylation of MCL1. We first examined effects of erlotinib versus MARCH5 depletion on MARCH5 substrates. Treatment with erlotinib again increased NOXA and decreased MCL1, but did not decrease other MARCH5 substrates (MiD49, MFN1, and FUNDC1) (*Figure 4D*). Interestingly, while MiD49 was increased in the MARCH5 knockout cells (see also *Figure 3E and H*), MFN1 and FUNDC1 were not altered, indicating that these latter substrates are not undergoing MARCH5-mediated degradation under basal conditions. In any case, this result indicates that erlotinib is not generally enhancing MARCH5 activity.

We then asked whether erlotinib alters the mitochondrial localization of MARCH5, or of MCL1. Consistent with previous reports, cellular fractionation showed that MARCH5 was primarily located to mitochondria (*Figure 4E*). Treatment with erlotinib for 2 hr (prior to a substantial decrease in MCL1) did not change this localization of MARCH5. Moreover, it did not increase the mitochondrial localization of MCL1, BIM or NOXA, indicating that erlotinib-mediated MCL1 degradation is not through increased targeting of these latter proteins to mitochondria. Finally, MARCH5 depletion did not clearly alter the fraction of MCL1 associated with mitochondria.

As MARCH5 may be activated by mitochondrial stress, we also asked whether tyrosine kinase inhibition had acute effects on mitochondria that may alter MARCH5 function. To address this we examined mitochondrial respiration in response to erlotinib or lapatinib in LNCaP-derived C4-2 cells, which were more suitable for these studies as they had stronger attachment to the culture plate. Similar to LNCaP cells, treatment with erlotinib or lapatinib for 4 hr under conditions used for the Seahorse assays decreased MCL1 in C4-2 cells (*Figure 4F*). We then treated with erlotinib or lapatinib for 2 hr and assessed oxygen consumption. Neither erlotinib nor lapatinib changed maximal oxygen consumption rate (*Figure 4G,H*), suggesting that EGFR inhibition is not promoting functional damage to mitochondrial regarding ATP production. Intriguingly, erlotinib and lapatinib increased basal oxygen consumption (ATP linked respiration) (*Figure 4G,I*), indicating a shift from fermentation to increased oxidative phosphorylation. The precise basis for this metabolic adaptation, and whether it is linked to activation of a stress response, is not clear. In any case, these findings indicate that MARCH5 is not altered in response to erlotinib, and that its increased degradation of MCL1 is driven primarily by the increase in NOXA.

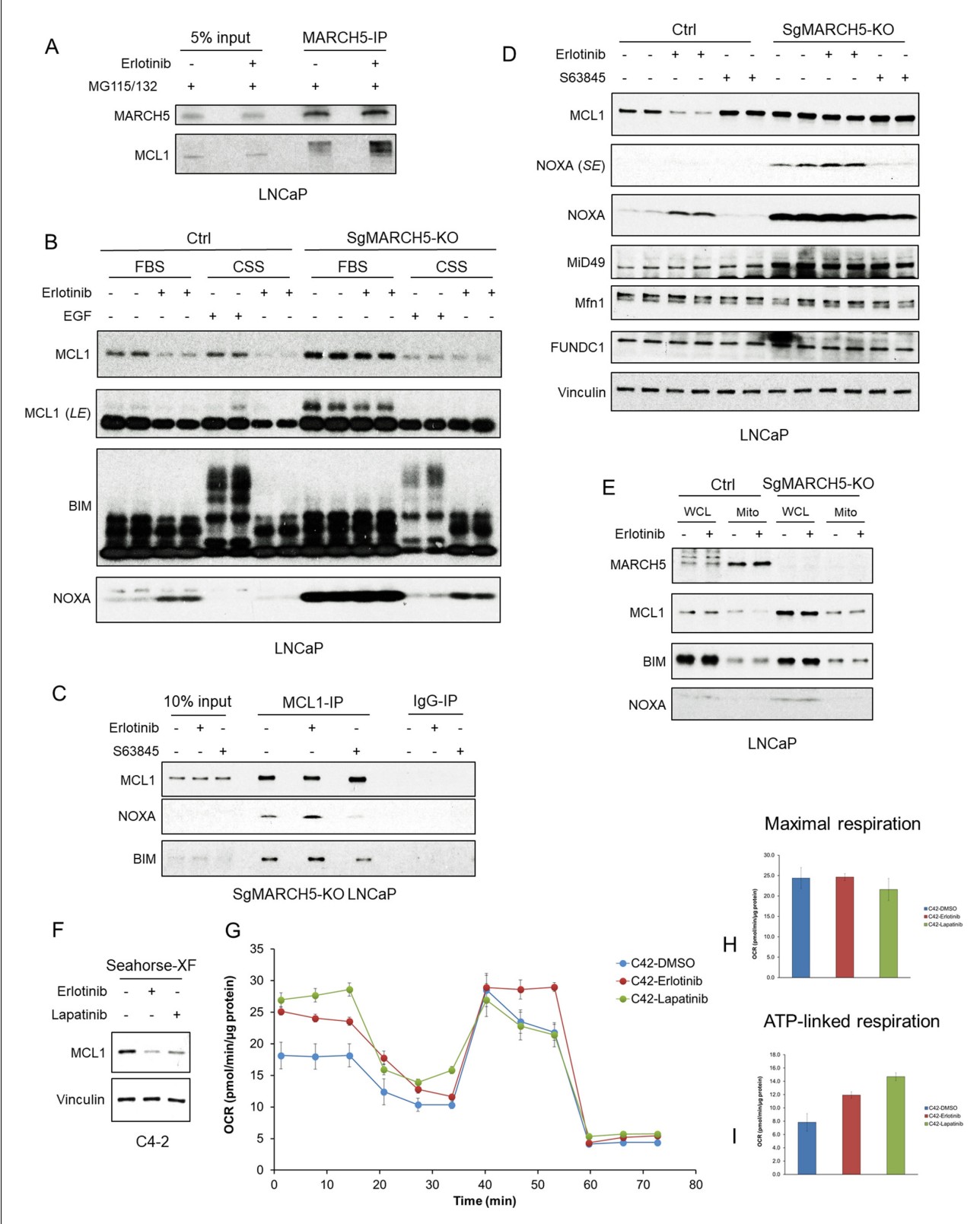

**Figure 4.** EGFR inhibition enhances MARCH5-MCL1 interaction without altering MARCH5 activity. (**A**) LNCaP cells were pretreated with proteasome inhibitors MG115 (10 µM) and MG132 (10 µM) for 1 hr, followed by treatment with erlotinib (10 µM) or DMSO for 3 hr. The cell lysates were subject to immunoprecipitation using anti-MARCH5 rabbit antibody with protein A agarose, followed by immunoblotting with anti-MARCH5 rabbit antibody or anti-MCL1 mouse antibody. The decreased mobility of MCL1 in the IP lanes may reflect impaired migration due to large amounts of Ig in the sample,

*Figure 4 continued on next page*

*Figure 4 continued*

but phosphorylation or other posttranslational modification is possible. (**B**) SgMARCH5-KO or control LNCaP cells were pre-incubated in normal serum medium (FBS) or charcoal-stripped serum medium (CSS) for 1 day, followed by treatment with erlotinib (10 µM) for 3 hr or EGF (100 ng/ml) for 30 min. LE, long exposure. (**C**) SgMARCH5-KO LNCaP cells were treated with erlotinib (10 µM), S63845 (0.5 µM), or DMSO for 3 hr. The cell lysates were immunopurified with anti-MCL1 rabbit antibody or control rabbit IgG and protein A agarose, followed by immunoblotting with mouse antibodies targeting for indicated proteins. (**D**) SgMARCH5-KO or control LNCaP cells were treated with erlotinib (10 µM), S63845 (500 nM), or DMSO for 3 hr. SE, short exposure. (**E**) SgMARCH5-KO or control LNCaP cells were treated with erlotinib (10 µM) for 2 hr. Proteins extracted from whole cell lysates (WCL) or isolated mitochondria (Mito) were analyzed by western blot. WCL, whole cell lysate. Mito, mitochondrial fraction. (**F**) LNCaP-derived C4-2 cells were incubated in Seahorse XF medium and treated with erlotinib (10 µM) or lapatinib (10 µM) for 4 hr. (**G–I**) C4-2 cells were treated with erlotinib (10 µM), lapatinib (10 µM), or DMSO for 3 hr, and maximal oxygen consumption rate (**H**) and ATP-linked oxygen consumption rate (**I**) were analyzed by a mitochondria stress test (**G**). Data in G-I are mean and standard deviation from three independent experiments. Immunoblot in (**B**) is representative of results obtained in two independent experiments, and the remainder are representative of at least three independent experiments.

The online version of this article includes the following figure supplement(s) for figure 4:

**Figure supplement 1.** MARCH5 depletion increases NOXA protein.

## Mitochondria-targeted agents can increase MCL1 degradation by MARCH5-dependent mechanism

MARCH5 regulates mitochondrial fission and fusion in response to mitochondrial stress (*Chen et al., 2017*; *Park et al., 2014*; *Yonashiro et al., 2006*; *Xu et al., 2016*; *Cherok et al., 2017*), suggesting that agents that alter mitochondria functions may enhance MARCH5-mediated degradation of MCL1 by a distinct mechanism. To assess this hypothesis, we examined the effects of a series of mitochondria-targeted agents on MCL1. Actinonin is an inhibitor of the human mitochondrial peptide deformylase that blocks mitochondrial protein translation (*Escobar-Alvarez et al., 2010*). Four-hour treatment with actinonin decreased MCL1 in LNCaP cells (*Figure 5A*). However, it also increased NOXA, suggesting that it may be acting similarly to tyrosine kinase inhibitors through an ISR, rather than by directly through MARCH5. Gamitrinib-TPP is a mitochondrial HSP90 inhibitor and can induce MCL1 degradation in glioblastoma cells (*Kang et al., 2010*). Consistent with previous data (*Karpel-Massler et al., 2017*; *Ishida et al., 2017*), gamitrinib-TPP rapidly decreased MCL1 in LNCaP cells, and this was also associated with an increase in NOXA (*Figure 5B*). The pyruvate dehydrogenase/α-ketoglutarate dehydrogenase inhibitor CPI-613 is another clinically promising agent that targets mitochondria (*Pardee et al., 2014*). Similar to actinonin and gamitrinib-TPP, treatment with CPI-613 decreased MCL1 and also increased NOXA (*Figure 5C*).

Significantly, each of these mitochondria-targeted agents increased ATF4 (*Figure 5C*), indicating an ISR mechanism for increasing NOXA. Consistent with this finding, and with a previous report on gamitrinib-TPP (*Karpel-Massler et al., 2017*), treatment with ISRIB impaired the upregulation of ATF4 and NOXA, and the reduction of MCL1, by each of these mitochondria-targeted agents (*Figure 5C*). Moreover, depleting NOXA with siRNA prevented the decrease in MCL1 in response to each of these agents (*Figure 5D*). Together these findings indicated that the increased MCL1 degradation in response to these agents was being driven by increased NOXA downstream of an ISR.

As further evidence for this conclusion, we found that the decrease in MCL1 by these mitochondria-targeted agents was proteasome-dependent, and was not associated with an increase in p53 (*Figure 5E,F*). Finally, we used a caspase inhibitor to confirm that these mitochondrial-targeted agents were not increasing MCL1 degradation through release and activation of caspases, which can degrade MCL1 (*Figure 5F*; *Figure 5—figure supplement 1A*). As a positive control for caspase inhibition, we showed that Z-DEVD-FMK could prevent caspase cleavage in response to erlotinib in combination with ABT-737 (*Figure 5F*; *Figure 5—figure supplement 1A*).

We next used siRNA to determine whether MCL1 degradation in response to these mitochondrial-targeted agents was mediated by MARCH5. Depleting MARCH5 markedly increased MCL1 and prevented the MCL1 loss in response to erlotinib and actinonin, although the effects of gamitrinib-TPP and CPI-613 were only partially suppressed, suggesting other ubiquitin ligases may contribute to this MCL1 degradation and partially compensate for the loss of MARCH5 (*Figure 5G*). Indeed, depletion of HUWE1 (which more modestly increased MCL1) partially impaired the effects of CPI-613 (*Figure 5H*). Finally, as expected and consistent with a previous study of gamitrinib-TPP (*Karpel-Massler et al., 2017*), MCL1 degradation by actinonin in combination with BCLXL/BCL2 inhibition by ABT-263 caused dramatic apoptosis in LNCaP cells (*Figure 5—figure supplement 1B*).

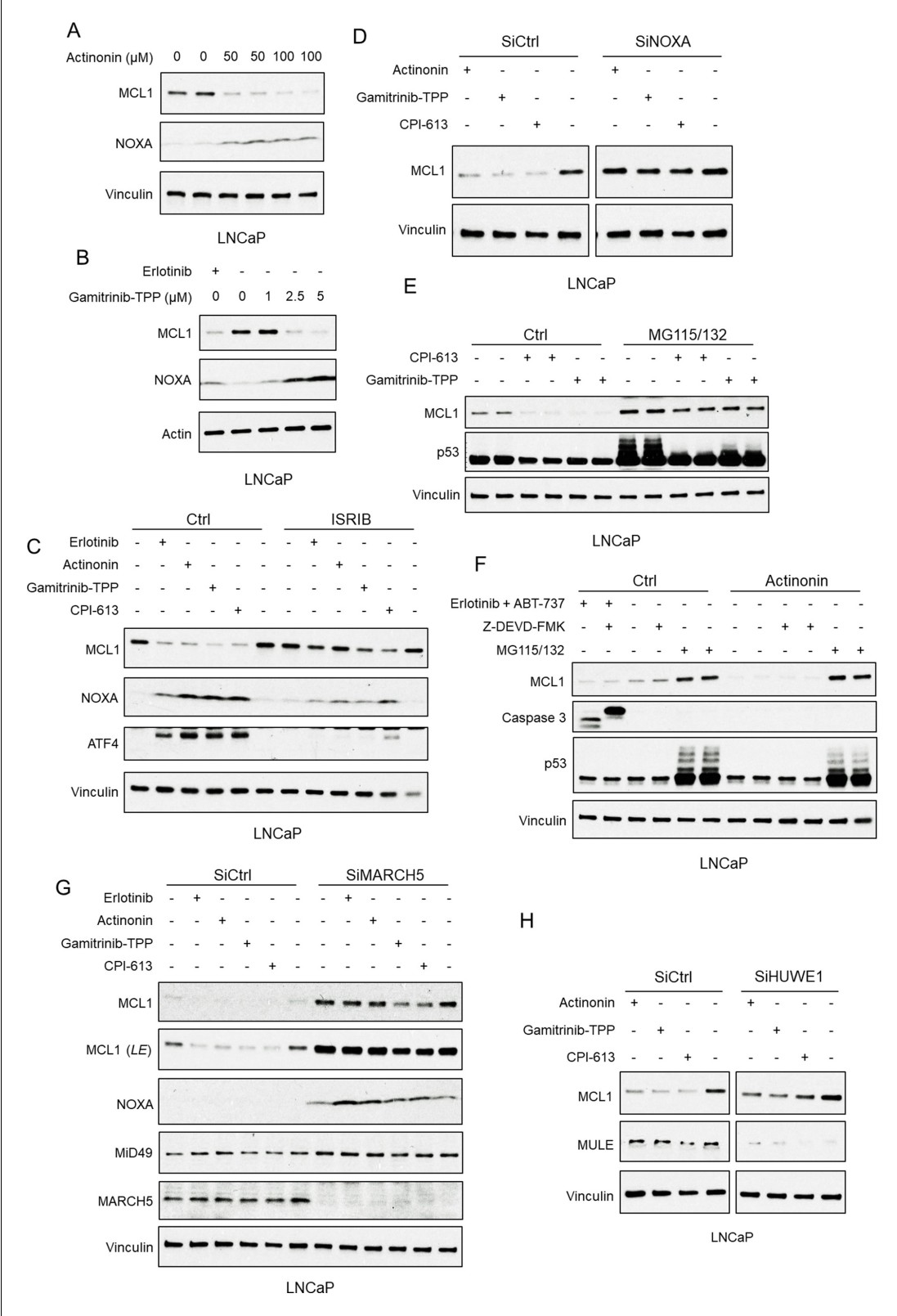

**Figure 5.** Mitochondria-targeted agents upregulate NOXA and induce MARCH5-dependent MCL1 degradation. (**A**) LNCaP cells were treated with human mitochondrial translation inhibitor actinonin (0–100 µM) for 5 hr, and whole cell lysates were then assessed by immunoblotting. (**B**) LNCaP cells were treated with mitochondrial HSP90 inhibitor gamitrinib-TPP (0–5 µM) or erlotinib (10 µM) for 9 hr. (**C**) LNCaP cells were pretreated with ISRIB trans-isomer (1 µM) for 1 hr, followed by treatment with erlotinib (10 µM), actinonin (100 µM), gamitrinib-TPP (5 µM), or pyruvate dehydrogenase/α-

*Figure 5 continued on next page*

Figure 5 continued

ketoglutarate dehydrogenase inhibitor CPI-613 (200 µM) for 5 hr. (D) LNCaP cells transfected with pooled NOXA siRNAs or non-target control siRNA were treated with actinonin (100 µM), gamitrinib-TPP (5 µM), CPI-613 (200 µM), or DMSO for 5 hr. (E) LNCaP cells were pretreated with MG115 (10 µM) and MG132 (10 µM) for 1 hr, followed by treatment with CPI-613 (200 µM), gamitrinib-TPP (5 µM), or DMSO for 4 hr. Efficacy of proteasome block by MG115/MG132 was confirmed by blotting for p53. (F) LNCaP cells were pretreated with MG115/MG132 (10 µM each), caspase inhibitor Z-DEVD-FMK (20 µM), or DMSO for 1 hr, followed by treatment with actinonin (75 µM), combination of erlotinib (10 µM) and BCLXL/BCL2 inhibitor ABT-737 (5 µM), or DMSO for 5 hr. Efficacy of caspase block by Z-DEVD-FMK and proteasome block by MG115/MG132 were confirmed by blotting for cleaved caspase three and p53, respectively. (G) LNCaP cells transfected with pooled MARCH5 siRNAs or non-target control siRNA were treated with erlotinib (10 µM), actinonin (100 µM), gamitrinib-TPP (5 µM), CPI-613 (200 µM), or DMSO for 5 hr. LE, long exposure. (H) LNCaP cells transfected with pooled HUWE1 (MULE) siRNAs or non-target control siRNA were treated with actinonin (100 µM), gamitrinib-TPP (5 µM), CPI-613 (200 µM), or DMSO for 5 hr. Immunoblots in (A and F) are representative of results obtained in two independent experiments, and the remainder are representative of at least three independent experiments.

The online version of this article includes the following figure supplement(s) for figure 5:

**Figure supplement 1.** Mitochondria-targeted agents increase caspase-independent MCL1 degradation and synergize with BCLXL/BCL2 inhibitor to induce apoptosis.

Overall, these results indicate that mitochondrial stress, similarly to kinase inhibitors, increases MCL1 degradation primarily through ISR-mediated activation and NOXA-dependent MARCH5-mediated ubiquitylation.

## MARCH5 genomic loss in PCa

Consistent with its antiapoptotic and hence oncogenic function, the *MCL1* gene is frequently amplified in multiple cancers (~10% in the largest reported PCa dataset) (*Figure 6A*). To determine whether MARCH5 may have tumor suppressor functions in vivo, we examined whether it had genomic alterations in PCa. Deep deletions of *MARCH5* were identified in up to ~5% of PCa across a series of data sets (*Figure 6B*), and *MARCH5* deletions (either shallow or deep deletion) are associated with shorter progression free survival (*Figure 6—figure supplement 1A*). In contrast, *HUWE1* loss was very rare (*Figure 6C*). Interestingly, assessing genomic alterations across cancers, *MARCH5* loss appears to be most common in PCa (*Figure 6D*). Significantly, this may reflect its genomic location adjacent to *PTEN* at 10q23, and hence co-deletion with *PTEN*. Indeed, in the TCGA primary PCa dataset, all cases with deep deletion of *MARCH5* also have *PTEN* deletion (*Figure 6E*). In contrast, *MARCH5* deletion appears to be occurring independently of *PTEN* loss in a subset of metastatic PCa.

*MCL1* amplification and *MARCH5* loss generally occur in distinct tumors, although their mutual exclusivity is not statistically significant (*Figure 6F*; *Figure 6—figure supplement 1B,C*). Relative to *MARCH5* and *MCL1*, oncogenic alterations in the genes encoding NOXA (*PMAIP1*) and BIM (*BCL2L11*) are rare (*Figure 6F*; *Figure 6—figure supplement 1B,C*). Finally, shallow deletions of *MARCH5*, suggesting single copy losses, appear to be relatively common in PCa, with a higher frequency in metastatic castration-resistant PCa versus primary PCa (*Figure 6G,H*; *Figure 6—figure supplement 1D,E*). Together these results support a tumor suppressor function for MARCH5, which may be related to its negative regulation of MCL1.

## MARCH5 loss decreases dependence on MCL1

The increased MCL1 in MARCH5 depleted cells suggested that these cells may have an increased dependence on MCL1. To assess effects of *MARCH5* loss on responses to MCL1 antagonists, we treated parental versus *MARCH5* knockout cells with S63845. As expected, both NOXA and BIM were markedly increased in the *MARCH5* knockout cells (*Figure 7A*). S63845 at the lowest concentration examined (1 µM) both stabilized MCL1 and decreased NOXA and BIM, consistent with S63845 binding to MCL1 and displacing NOXA and BIM, and with their subsequent increased degradation. Surprisingly, despite the apparent substantial displacement of NOXA and BIM from MCL1, the increase in apoptosis (as assessed by cleaved caspase 3, CC3, and cleaved PARP, cPARP) was only observed at the highest concentration of drug (20 µM). Identical results were obtained with a second MCL1 antagonist (AZD5991) (*Figure 7B*). We also examined cells stably overexpressing ectopic MCL1. These cells similarly had marked increases in NOXA and BIM, which were decreased

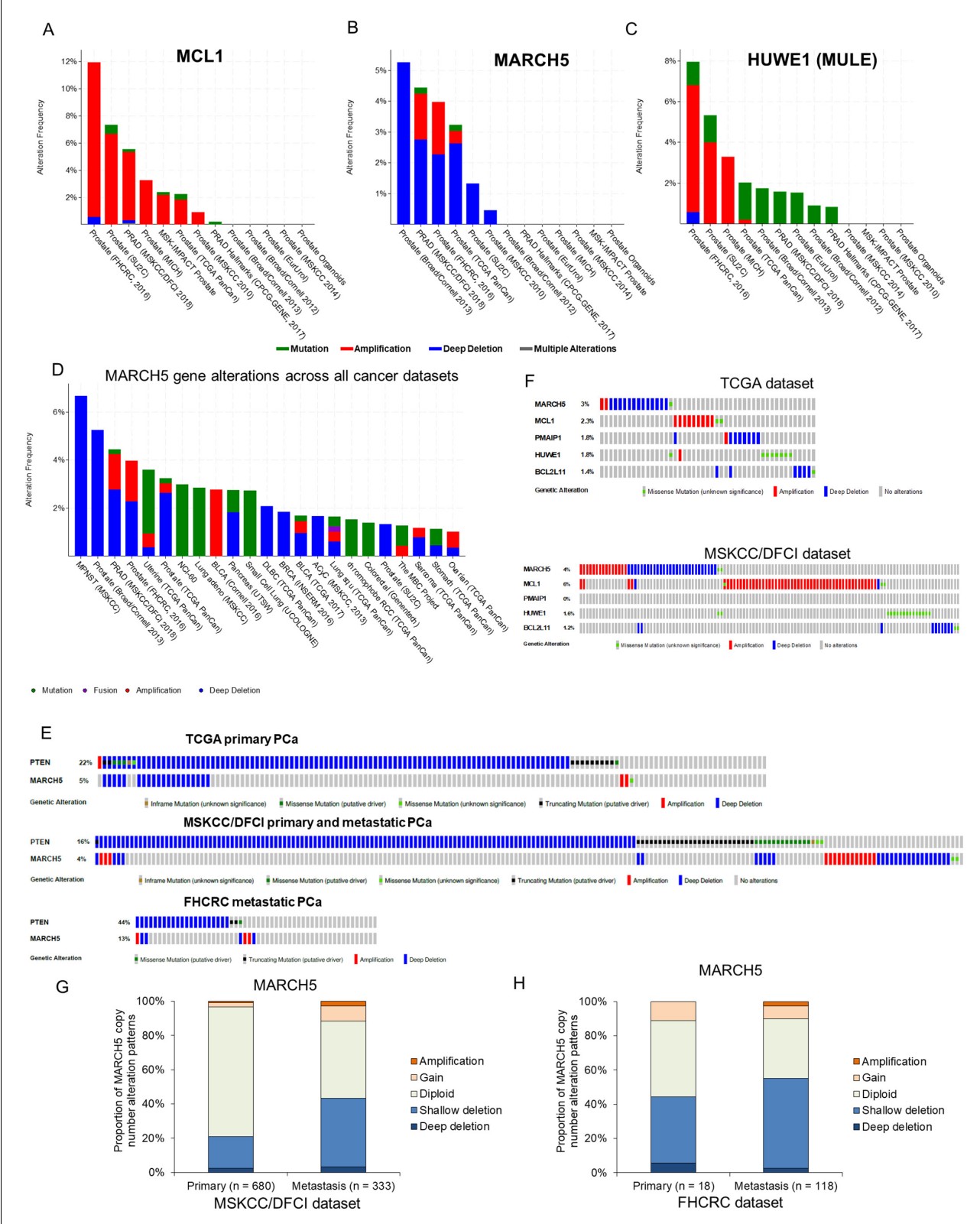

**Figure 6.** MARCH5 deletion or MCL1 amplification exists in subsets of PCa patients. (**A–C**) Molecular profiles (copy number alterations and mutation) of *MCL1* (**A**), *MARCH5* (**B**), and *HUWE1* (MULE) (**C**) among PCa datasets in the cBioPortal for Cancer Genomics (http://cbioportal.org). (**D**) Frequency and patterns for *MARCH5* gene alterations across all cancer datasets, frequency in MPNST (malignant peripheral nerve sheath tumors) reflects only one case. (**E**) Overlap between genomic alterations in *MARCH5* and *PTEN*. (**F**) Gene alterations for *MARCH5*, *MCL1*, *PMAIP* (NOXA), *HUWE1* (MULE), and
*Figure 6 continued on next page*

*Figure 6 continued*

*BCL2L11* (BIM) in TCGA dataset and MSKCC/DFCI PCa datasets. (**G and H**) Proportion of copy number alteration patterns for *MARCH5* between primary prostate tumor and metastatic prostate tumor samples in MSKCC/DFCI dataset (**G**) and FHCRC dataset (**H**).

The online version of this article includes the following figure supplement(s) for figure 6:

**Figure supplement 1.** MARCH5 deletion is observed in subsets of PCa patients.

---

in response to 1 µM S63845 (*Figure 7C*) or AZD5991 (*Figure 7D*), but apoptotic responses again required high drug concentrations.

Although the MARCH5 depleted and MCL1 overexpressing cells showed increased apoptosis in response to MCL1 antagonists, it was at substantially higher drug concentrations than those needed for release of BIM and NOXA, suggesting that it may be an off-target effect. One factor contributing to the lack of efficacy at lower concentrations may be that the BIM and NOXA that is displaced from MCL1 by S63845 and AZD5991 appears to undergo rapid degradation, as their levels in the treated MARCH5 depleted or MCL1 overexpressing cells were not markedly higher than in the parental control cells (*Figure 7A–D*). Another factor may be that the high levels of NOXA and BIM in the MARCH5 depleted cells and MCL1 overexpressing cells are effectively competing with BAK for MCL1 binding, so that these cells are less dependent on MCL1 (and more dependent on other anti-apoptotic BCL2 family proteins) to buffer BAK. However, arguing against this latter mechanism, by coimmunoprecipitation we found that MCL1 was binding increased levels of BAK, as well as NOXA and BIM, in the MARCH5 depleted cells and the MCL1 overexpressing cells (*Figure 7E*). Alternatively, as MCL1 has a preference for binding BAK versus BAX (*Llambi et al., 2011*), it is possible that the increased levels of MCL1 are adequate to neutralize BAK even at drug concentrations up to 10 µM, and that the effects at 20 µM are on-target. In support of this latter mechanism, we found that BAK was not increased in the *MARCH5* knockout cells (*Figure 7F*), which may allow the high levels of MCL1 to effectively buffer BAK despite treatment with S63845 or AZD5991.

In contrast to BAK, in the unactivated state BAX is localized primarily in the cytoplasm and may be buffered mostly by BCLXL and BCL2. Significantly, BAX protein expression was decreased in the *MARCH5* knockout cells (*Figure 7F*). The decrease in BAX after *MARCH5* loss (as well as the decrease in PUMA) suggested that the *MARCH5* knockout cells may have decreased capacity to buffer BAX and be very sensitive to acute increases in free BAX, and hence be more dependent on BCL2 or BCLXL. Therefore, we assessed responses to the BCL2/BCLXL antagonist ABT-263 (navito-clax). Significantly, ABT-263 treatment caused a marked apoptotic response specifically in the *MARCH5* knockout cells (*Figure 7G*). As we reported previously (*Arai et al., 2018*), ABT-263 could induce apoptosis in control parental cells in combination with S63845, but the addition of S63845 only minimally enhanced apoptosis in the ABT-263 treated *MARCH5* knockout cells (*Figure 7H*). The BCL2-specific antagonist ABT-199 (venetoclax) was not effective, indicating that the efficacy of ABT-263 is due to BCLXL inhibition (*Figure 7I*).

Of note, a previous study similarly found that *MARCH5* knockdown could increase MCL1 and sensitize to BCLXL inhibition, and suggested that increased NOXA was suppressing the antiapoptotic activity of MCL1 (47). While this increased NOXA may be a factor, our data indicate that the increased MCL1 in *MARCH5* knockdown cells is sequestering substantial levels of both BAK and BIM (see *Figure 7E*). To explore other mechanisms, we examined the Avana CRISPR screen dataset through the Broad DepMap site (https://depmap.org) to identify cell lines that were dependent on *MARCH5* and genes that have most similar patterns of dependency (*Meyers et al., 2017*). Interestingly, the gene that was most co-dependent with *MARCH5* was *MCL1* (*Figure 7J,K*). Conversely, the gene most co-dependent with MCL1 was MARCH5. This strong co-dependency was also observed in screens with another CRISPR library (*Figure 7—figure supplement 1A,B,C*). Based on these results and our data, we suggest that MARCH5, while acting as a ubiquitin ligase for NOXA-liganded MCL1, may also have a distinct function in conjunction with MCL1 to suppress mitochondrial membrane permeabilization by BAX.

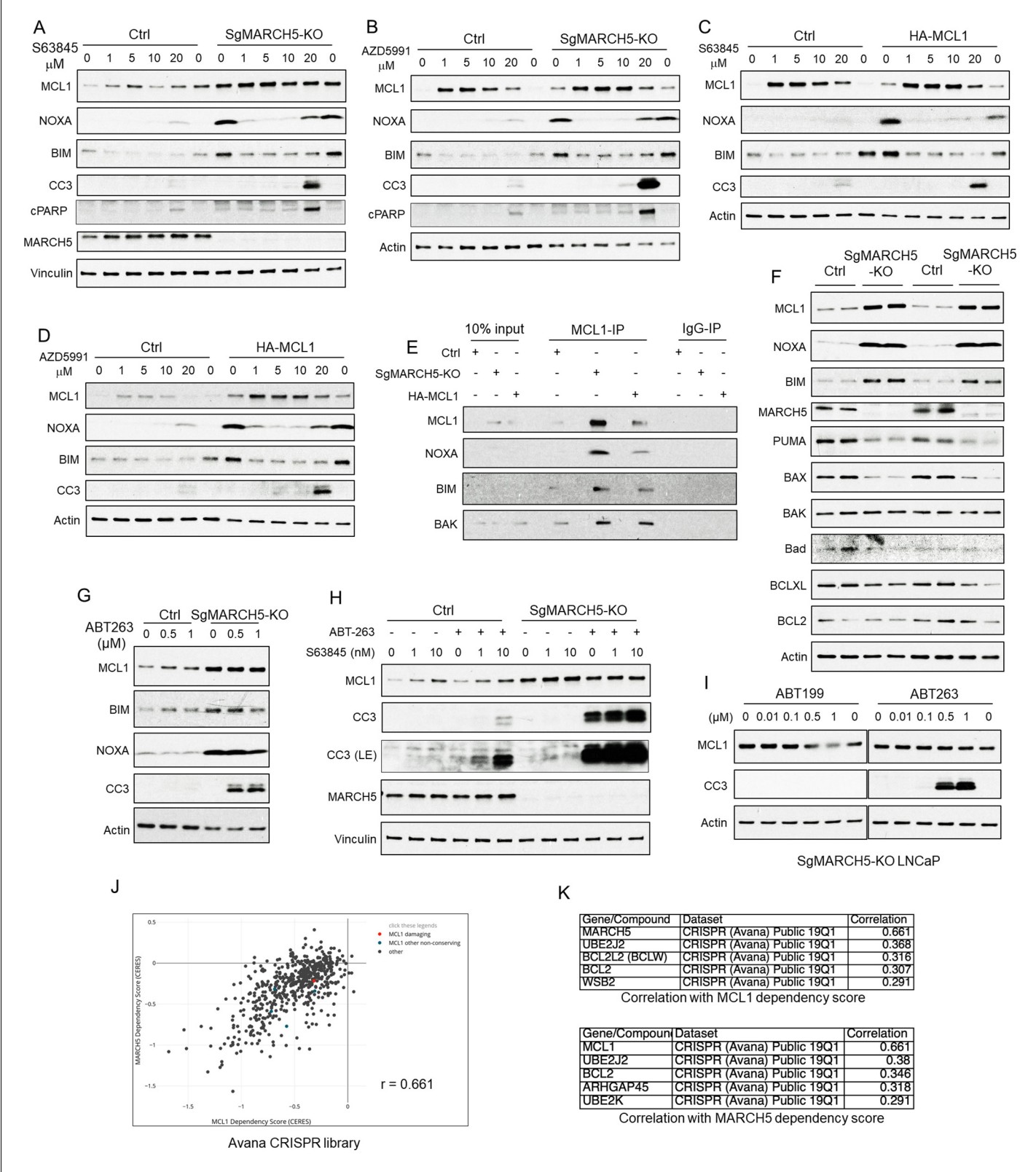

**Figure 7.** MARCH5 depletion sensitizes BH3 mimetics to drive apoptosis in PCa cells. (**A and B**) SgMARCH5-KO or control LNCaP cells were treated with S63845 (0–20 µM) (**A**) or another MCL1 inhibitor AZD5991 (0–20 µM) (**B**) for 12 hr. Apoptosis induction was detected with cleaved caspase 3 (CC3) and cleaved PARP (cPARP) signals. (**C and D**) HA-MCL1 or control LNCaP cells were treated with S63845 (0–20 µM) (**C**) or AZD5991 (0–20 µM) (**D**) for 12 hr. (**E**) Cell lysates of SgMARCH5-KO, HA-MCL1, or control LNCaP with same protein amounts were immunoprecipitated using anti-MCL1 mouse

*Figure 7 continued on next page*

*Figure 7 continued*

antibody or control mouse IgG with protein G agarose, followed by immunoblotting with rabbit antibodies targeting MCL1, BIM, or BAK, or mouse antibody targeting NOXA. (F) SgMARCH5-KO or control LNCaP cells (biological replicates) were lysed and immunoblotted for indicated proteins. (G) SgMARCH5-KO or control LNCaP cells were treated with BCL2/BCLXL inhibitor ABT-263 (0–1 μM) for 9 hr. (H) sgMARCH5-KO or control LNCaP cells were treated with S63845 (0–10 nM) and ABT-263 (500 nM) or DMSO and for 9 hr. LE, long exposure. (I) SgMARCH5-KO LNCaP cells were treated with BCL2 inhibitor ABT-199 (0–1 μM) or ABT-263 (0–1 μM) for 9 hr. (J) Correlation between *MCL1* dependency score and *MARCH5* dependency score in AVANA CRISPR screen. (K) Lists of top five genes whose dependency scores are correlated with *MCL1* dependency score (upper) or *MARCH5* dependency score (lower). Immunoblot in (F) is representative of results obtained in two independent experiments, and the remainder are representative of at least three independent experiments.

The online version of this article includes the following figure supplement(s) for figure 7:

**Figure supplement 1.** MARCH5 shows codependency with MCL1 in DepMap CRISPR-CAS9 essentiality screens in cancer cells.

# Discussion

We reported previously that treatment with several kinase inhibitors could markedly increase MCL1 degradation, and that this increase was not mediated by well-established MCL1 ubiquitin ligases including βTRCP, FBW7, HUWE1 (22). In this study we initially found that erlotinib treatment rapidly increased expression of NOXA, and that the increased MCL1 degradation was NOXA-dependent. We subsequently found that the increase in NOXA was driven by ISR activation, with subsequent increase in ATF4 protein and NOXA transcription. Previous studies have shown that NOXA binding can increase the degradation of MCL1 (39), and have implicated the ubiquitin ligases HUWE1 or PARKIN in this degradation (*Gomez-Bougie et al., 2011*; *Zhong et al., 2005*; *Warr et al., 2005*; *Carroll et al., 2014*). Here, we identified the mitochondria-associated ubiquitin ligase MARCH5 as the primary mediator of this kinase inhibitor initiated NOXA-dependent MCL1 degradation. Significantly, MARCH5 depletion both abrogated the decrease in MCL1 in response to erlotinib and substantially increased basal MCL1 in multiple prostate, breast, and lung cancer cell lines, indicating that MARCH5 makes a major contribution to regulating basal levels of MCL1. The physiological significance of *MARCH5* as a tumor suppressor gene through regulation of MCL1 is further supported by its genomic loss in a subset of cancers. Importantly, MARCH5 depleted cells, which have increased levels of both MCL1 and NOXA, have increased sensitivity to MCL1 antagonists (although at high concentrations that may have off-target effects) and to the BH3 mimetic drug navitoclax (due to targeting BCLXL), suggesting therapeutic approaches for MARCH5-deficient tumors.

The ISR with increased translation of ATF4 can be driven by multiple stimuli that converge on phosphorylation of eIF2α, with subsequent increased translation of ATF4 and increased expression of many genes that can contribute to resolving metabolic stress or driving apoptosis. Importantly, the precise downstream consequences of ISR activation are context dependent, but apoptosis is usually induced after prolonged stress and mediated by ATF4 induction of CHOP (*Corazzari et al., 2017*; *Iurlaro and Muñoz-Pinedo, 2016*). However, ATF4 has been reported to directly upregulate the *PMAIP1* gene (encoding NOXA) (*Armstrong et al., 2010*; *Wang et al., 2009*), which would be consistent with the rapid time course of NOXA induction that correlated with increased ATF4. The prominence of this ATF4 induction of NOXA in response to receptor tyrosine kinase inhibitors may reflect interactions between multiple pathways downstream of these receptors. However, as these kinase inhibitors are ATP analogues, we cannot rule out off target effects on some ATP-dependent processes. Indeed, treatment with erlotinib or lapatinib rapidly increased basal oxygen consumption, indicating a shift from fermentation to oxidative phosphorylation to increase ATP synthesis, and a metabolic stress that may contribute to ISR activation. Moreover, ISR stimulation in this case appears to be mediated by both PERK and GCN2, consistent with a broad metabolic stress affecting several cellular processes.

NOXA binding to MCL1 appears to stabilize a conformation that can drive its interaction with ubiquitin ligases including HUWE1 and, as shown in this study, with MARCH5 (*Gomez-Bougie et al., 2011*; *Guikema et al., 2017*; *Song et al., 2016*). Indeed, the finding that MARCH5 depletion prevented the degradation of MCL1 in response to NOXA induction indicates that MARCH5 is the major ubiquitin ligase mediating NOXA-induced MCL1 degradation. We further found that MARCH5 depletion increased MCL1 in multiple cell lines, indicating that MARCH5 plays a substantial role in regulating MCL1 under basal conditions, although this may still be NOXA-dependent and could

reflect constitutive levels of stress in tumor cells. This latter result is consistent with previous data from two groups showing that that MARCH5 depletion can increase MCL1 (*Cherok et al., 2017*; *Subramanian et al., 2016*). Interestingly, and consistent with the latter study, we found that MARCH5 depletion was associated with an increase in NOXA. This increase in NOXA was not due to increased p53-mediated transcription. Instead, it reflects NOXA stabilization by MCL1 binding, as NOXA levels decreased rapidly when NOXA was competed off with an MCL1 antagonist. However, we cannot rule out the possibility that MARCH5 also indirectly regulates NOXA levels by coupling the degradation of MCL1 in MCL1-NOXA complexes to the degradation of NOXA. Finally, a very recent study found that MARCH5 is a major mediator of MCL1 and NOXA degradation in response to taxane-mediated mitotic arrest (*Haschka et al., 2020*).

As MARCH5 is located on the mitochondrial outer membrane, we further asked whether its degradation of MCL1 might be enhanced by drugs that perturb mitochondrial function. Indeed, we found that all three agents examined (actinonin, gamitrinib-TPP, and CPI-613) caused a MARCH5-dependent increase in MCL1 degradation. However, this did not appear to reflect a direct effect on MARCH5. It was instead associated with a stress response, with increased ATF4 and NOXA, similarly to the response to kinase inhibitors. These findings are consistent with a previous study of gamitrinib-TPP that found this agent could activate a stress response with an increase in NOXA and decrease in MCL1 (*Karpel-Massler et al., 2017*). Further studies are needed to determine whether MARCH5-mediated degradation of MCL1 can be enhanced by additional agents that alter mitochondrial function through alternative mechanisms.

MCL1 is an inhibitor of apoptosis that acts by neutralizing BAK/BAX and by sequestering activators of BAK/BAX such as BIM, and by also sequestering the less potent activators NOXA and PUMA. Therefore, we anticipated that cells expressing high levels of MCL1 due to MARCH5 depletion or overexpression of ectopic MCL1 would have increased dependence on MCL1 to neutralize BAK/BAX and sequester BIM, NOXA, and PUMA. Indeed, we confirmed that MCL1 was binding increased levels of these proteins in MARCH5 knockout and MCL1 overexpressing cells, and that MCL1 antagonists could induce apoptosis in the MARCH5 knockout and MCL1 overexpressing cells, but not the control cells. However, while the apparent release of BIM and NOXA from MCL1 and their subsequent degradation were observed at relatively low concentrations of S63845 or AZD5991, the induction of apoptosis required ~20 µM of these drugs. This requirement for higher drug levels may reflect the very high levels of MCL1 and its subsequent persistent engagement of BAK, despite treatment with MCL1 antagonists.

The MARCH5 knockout cells also underwent apoptosis in response to BCLXL antagonism with ABT-263, while apoptosis in the parental cells required antagonism of both BCLXL and MCL1. A previous study similarly found that MARCH5 depletion could sensitize to ABT-263, and suggested it may be due to high levels of NOXA that are antagonizing the antiapoptotic functions of MCL1 (*Subramanian et al., 2016*). However, we found that MCL1 in MARCH5 knockout cells was binding increased BAK and BIM, as well NOXA. Alternatively, as suggested by our data, there may be a codependency between MARCH5 and MCL1 for buffering of BAX, so that BCLXL in the MARCH5 knockout cells becomes critical to suppress the activity of BAX.

While more studies are clearly needed to further define how *MARCH5* loss (or *MCL1* amplification) alters responsiveness to BH3 mimetics, this study indicates that *MARCH5* loss, which appears to be relatively common in PCa, confers vulnerabilities to BH3 mimetic drugs. However, challenges to exploiting these vulnerabilities include thrombocytopenia caused by BCLXL inhibition, and the possible requirement for high concentrations of MCL1 antagonists, whose toxicity profile remains to be established. Importantly, the available MCL1 antagonists are all noncovalent and stabilize MCL1, which may limit their ability to abrogate MCL1 interaction with BAK. Therefore, it is possible that antagonists that drive MCL1 degradation, possibly by mimicking the NOXA BH3 domain, might be more potent and effective. Finally, approaches that selectively cause robust ISR activation in tumor cells, with increased NOXA and MCL1 degradation, may create an exploitable therapeutic window for BCLXL antagonists.

## Materials and methods

### Key resources
Key resources are listed in Key Resources Table (*Supplementary file 1*).

### Cell culture
LNCaP, C4-2, PC3 and RV1 cells were cultured in RPM1640 medium (#MT10040CV, Fisher Scientific) with 10% FBS (#26140079, Fisher Scientific) and penicillin-streptomycin (100 IU/ml) (#15140122, Fisher Scientific). DU145, MDA-MB-468, MCF7, and A549 cells were cultured in DMEM medium (#MT10013CV, Fisher Scientific) with 10% FBS and penicillin-streptomycin (100 IU/ml). All cells were obtained from ATCC. Cell identity was confirmed by STR analysis, and Mycoplasma testing was negative. For most immunoblotting or quantitative RT-PCR experiments, cells were grown to around 50% confluence for 1 day and then treated with indicated drugs. Transient transfections for HA-tagged MARCH5 plasmid (#HG21559-NY, Sino Biological) were carried out using Lipofectamine 3000 (#L3000075, Fisher Scientific) following the manufacturer's instruction. Erlotinib (#S7786), lapatinib (#S2111), dinaciclib (#S2768), cabozantinib (#S1119), ABT-263 (#S1001), ABT-737 (#S1002), and ABT-199 (#S8048) were from Selleck Chemicals. Gamitrinib-TPP was kindly provided by Dr. Dario Altieri (The Wistar Institute). AZD5991 was provided AstraZeneca. S63845 (#HY-100741), actinomycin D (#HY-17559), ISRIB trans-isomer (#HY-12495), MLN4924 (#HY-70062), MG-132 (#HY-13259), CPI-613 (#HY-15453), and Z-DEVD-FMK (#HY-12466) were from MedChem Express. MG-115 (#C6706), actinonin (#A6671), and epidermal growth factor (EGF) (#E9644) were from Sigma-Aldrich.

### Immunoblotting
Cells were lysed in RIPA buffer (#PI89900, Fisher Scientific) supplemented with protease inhibitor (#PI78437, Fisher Scientific) and phosphatase inhibitor cocktails (#PI78426, Fisher Scientific). Blots were incubated with rabbit anti-ATF4 (rabbit, 1:2000) (#ab184909, Abcam), anti-Bad (rabbit, 1:500) (#9239, Cell Signaling Technology), anti-BAK (rabbit, 1:1000) (#12105, Cell Signaling Technology), anti-BAX (rabbit, 1:1000) (#5023, Cell Signaling Technology), anti-β-actin (mouse, 1:10000) (#ab6276, Abcam), anti-BCL2 (rabbit, 1:500) (#4223, Cell Signaling Technology), anti-BCLXL (rabbit, 1:1000) (#2764, Cell Signaling Technology), anti-BIM (rabbit, 1:1000) (#2933, Cell Signaling Technology), anti-BIM (mouse, 1:500) (#sc-374358, Santa Cruz Biotechnology), anti-cleaved caspase 3 (CC3) (rabbit, 1:250) (#9664, Cell Signaling Technology), anti-FUNDC1 (rabbit, 1:1000) (#PA5-48853, Fisher Scientific), anti-HA (rabbit, 1:1000) (#3724, Cell Signaling Technology), anti-MARCH5 (rabbit, 1:2000) (#06–1036, EMD Millipore), anti-MCL1 (rabbit, 1:1000) (#5453, Cell Signaling Technology), anti-MCL1 (mouse, 1:1000) (#sc-12756, Santa Cruz Biotechnology), anti-Mfn1 (mouse, 1:1000) (#sc-166644, Santa Cruz Biotechnology), anti-MiD49 (SMCR7) (rabbit, 1:1000) (#SAB2700654, Sigma Aldrich), anti-MULE (HUWE1) (mouse, 1:500) (#5695, Cell Signaling Technology), anti-NOXA (mouse, 1:250) (#ab13654, Abcam), anti-p27 (rabbit, 1:1000) (#3686, Cell Signaling Technology), anti-p53 (mouse, 1:1000) (#sc-126, Santa Cruz Biotechnology), anti-p62 (rabbit, 1:1000) (#5114, Cell Signaling Technology), anti-PARP (rabbit, 1:1000) (#9532, Cell Signaling Technology), anti-phospho-eIF2$\alpha$ Ser51 (rabbit, 1:1000) (#9721, Cell Signaling Technology), anti-PUMA (rabbit, 1:500) (#12450, Cell Signaling Technology), or anti-vinculin (mouse, 1:20000) (#sc-73614, Santa Cruz Biotechnology), and then with 1:5000 of anti-rabbit (#W401B) or anti-mouse (#W402B) secondary antibodies (Promega).

### RT-PCR
Quantitative real-time RT-PCR (qRT-PCR) amplification was performed on RNA extracted from cells using RNeasy Mini kit (#74104, Qiagen). RNA (50 ng) was used for each reaction, and the results were normalized by co-amplification of *18S ribosomal RNA (rRNA)* or *GAPDH*. Reactions were performed on an ABI Prism 7700 Sequence Detection System (Thermo Fisher Scientific) using TaqMan one-step RT-PCR reagents (#4444434, Fisher Scientific). Primer mix for *MARCH5* (Hs00215155_m1), *MCL1* (Hs01050896_m1), NOXA (*PMAIP*, Hs00560402_m1), *18S rRNA* (#4319413E), and *GAPDH* (#4326317E) was purchased from Thermo Fisher Scientific.

## RNA interference

For transient silencing of target genes, cells were transfected with pooled Bim siRNAs (#L-004383-00-0005, Dharmacon), an individual MARCH5 siRNA (s29332, Fisher), pooled MARCH5 siRNAs (#L-007001-00-0005, Dharmacon), pooled MULE (HUWE1) siRNAs (#L-007185-00-0005, Dharmacon), pooled NOXA siRNAs (#L-005275-00-0005, Dharmacon), three NOXA individual siRNAs (#s10708-10710, Thermo Fisher Scientific), pooled GCN2 (EIF2AK4) siRNAs (#L-005314-00-0005, Horizon Discovery), pooled PERK (EIF2AK3) siRNAs (#L-004883-00-0005, Horizon Discovery), or control non-target siRNA (#D-001810-01-05, Dharmacon) using Lipofectamine RNAiMAX (#13778150, Fisher Scientific) following the manufacturer's instruction. These transfected cells were then analyzed 48–72 hr later.

## Generation of cell line stably overexpressing HA-MCL1

LNCaP cells stably overexpressing HA-tagged MCL1 were previously generated (*Arai et al., 2018*). Briefly, LNCaP cells were transfected with HA-MCL1 (kindly provided by Dr. Wenyi Wei, BIDMC) using Lipofectamine 3000 and then selected with 750 µg/ml of G418 for two weeks.

## Generation of MARCH5 or MCL1 knockout cell line

LNCaP cells were co-transfected with MARCH5 CRISPR/Cas9 knockout (KO) plasmid (pool of 3 guide RNAs) (#sc-404655) and MARCH5 HDR plasmid (#sc-404655-HDR) at a ratio of 1:1 using Lipofectamine 3000. Cells were then selected with 2 µg/ml of puromycin for two weeks. The selective medium was replaced every 2–3 days. The single clones were picked and checked for MARCH5 expression. Control CRISPR/Cas9 plasmid (sc-418922) was used as a negative control. MCL1-KO LNCaP cells were previously generated (*Arai et al., 2018*) using MCL1 CRISPR/CAS9 KO plasmid (#sc-400079) and MCL1 HDR plasmid (#sc-400079-HDR). All plasmids were from Santa Cruz Biotechnology.

## Coimmunoprecipitation (Co-IP)

Control, MARCH5-KO, or HA-MCL1 LNCaP cells were treated with or without indicated drugs and were lysed in IP lysis buffer (#87788, Fisher Scientific) supplemented with protease and phosphatase inhibitor cocktails. The cell lysates were immunopurified with anti-MARCH5 rabbit, anti-MCL1 rabbit, anti-MCL1 mouse antibody, or control rabbit or mouse IgG overnight, and then were incubated with protein A or G agarose beads for 2 hr. The beads were washed five times with IP lysis buffer and were boiled for 5–10 min in two times Laemmli sample buffer (#1610737, Bio-Rad) with 2-mercaptoethanol (#BP176-100, Fisher Scientific). After centrifugation, the supernatants were immunoblotted for indicated proteins.

## Analysis of protein phosphorylation status

MARCH5-KO or control LNCaP cells were seeded in 10% FBS or 5% Charcoal Stripped Serum (CSS) medium for 1 day. These cells were treated with erlotinib for 3 hr or EGF for 30 min and were lysed in RIPA buffer supplemented with protease and phosphatase inhibitor cocktails. The cell lysates were immediately boiled for 5 min in laemmli sample buffer with 2-mercaptoethanol and were applied to SuperSep Phos-tag gel (#198–17981, FUJIFILM WAKO Chemicals), followed by immunoblotting for indicated proteins.

## Mitochondria isolation

Intact mitochondria were obtained from sgMARCH5-KO or control LNCaP cells with or without erlotinib treatment using mitochondria isolation kit (#89874, Thermo Fisher Scientific) following the manufacturer's instruction. Briefly, pelleted cells were incubated in mitochondria isolation reagents with protease and phosphatase inhibitors. After the incubation and centrifugation, the remaining pellets (isolated mitochondria) were used for western blot analysis.

## Seahorse analysis

C4-2 cells were seeded in 96-well plate in 10% FBS medium for 1 day. The medium was changed to Seahorse XF medium (#103576–100, Agilent Technologies) supplemented with 10 mM glucose (#103577–100, Agilent Technologies), 1 mM pyruvate (#103578–100, Agilent Technologies) and 2

mM glutamine (#103579–100, Agilent Technologies) before analysis. Real-time oxygen consumption rate (OCR) of these cells was measured using the Seahorse Extracellular Flux (XFe-96) analyzer (Agilent Technologies). Protein concentration of cells in each well was determined, and OCR value was normalized to µg/protein.

### Statistical analysis

Significance of difference between two groups was determined by two-tailed Student's t test using R software (version 3.3.2). Statistical significance was accepted at $p < 0.05$.

## Acknowledgements

We thanks Drs. Dario Altieri and Wenyi Wei for reagents, Dr. Mariusz Karbowski for helpful discussions, Dr. Xiaowen Liu for assistance with Seahorse studies, and Balk lab members for feedback.

## Additional information

### Funding

| Funder | Grant reference number | Author |
| --- | --- | --- |
| National Cancer Institute | P01 CA163227 | Steven P Balk |
| Congressionally Directed Medical Research Programs | W81XWH-16-1-0431 | Steven P Balk |
| JSPS | 20K09518 | Seiji Arai |
| Congressionally Directed Medical Research Programs | W81XWH-18-1-0379 | Steven P Balk |
| National Cancer Institute | P50CA090381 | Steven P Balk |

The funders had no role in study design, data collection and interpretation, or the decision to submit the work for publication.

### Author contributions

Seiji Arai, Conceptualization, Data curation, Formal analysis, Investigation, Methodology, Writing - original draft, Writing - review and editing; Andreas Varkaris, Conceptualization, Data curation, Formal analysis, Investigation, Project administration, Writing - review and editing; Mannan Nouri, Data curation, Investigation; Sen Chen, Conceptualization, Formal analysis, Investigation, Methodology; Lisha Xie, Formal analysis, Investigation; Steven P Balk, Conceptualization, Resources, Data curation, Formal analysis, Supervision, Funding acquisition, Validation, Investigation, Visualization, Methodology, Writing - original draft, Project administration, Writing - review and editing

### Author ORCIDs

Seiji Arai  https://orcid.org/0000-0002-9514-735X
Andreas Varkaris  https://orcid.org/0000-0001-5776-8844
Steven P Balk  https://orcid.org/0000-0002-4546-7371

### Decision letter and Author response

Decision letter https://doi.org/10.7554/eLife.54954.sa1
Author response https://doi.org/10.7554/eLife.54954.sa2

## Additional files

### Supplementary files

- Supplementary file 1. Key Resources Table.
- Transparent reporting form

Data availability
No new data sets are generated.

The following datasets were generated:

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
