## [Decision Letter]

**Acceptance summary:**

This work shows that MARCH5 promotes NOXA-dependent MCL1 degradation in response to kinase inhibitors downstream of the ISR, and synergizes with BH3 mimetics to kill tumor cells. This is a novel mechanism that has significance for cancer treatment.

**Decision letter after peer review:**

Thank you for submitting your article "Ubiquitin Ligase MARCH5 Regulates Apoptosis through Mediation of Stress-Induced and NOXA-Dependent MCL1 Degradation" for consideration by *eLife*. Your article has been reviewed by three peer reviewers, one of whom is a member of our Board of Reviewing Editors, and the evaluation has been overseen by Maureen Murphy as the Senior Editor. The reviewers have opted to remain anonymous.

The reviewers have discussed the reviews with one another and the Reviewing Editor has drafted this decision to help you prepare a revised submission.

Summary:

This work shows that MCL1 levels are reduced by treatment of PrCA cells with EGFR inhibitors in NOXA-dependent manner that requires the integrated stress response. Degradation of MCL1 in response to EGFRi was dependent on the MARCH5 E3 Ub ligase. Similarly, actinonin treatment to induce mitochondrial stress decreased MCL1. The authors also show that MARCH5 is lost in ~ 5% of primary PrCA and associated with poor prognosis and that MARCH5 loss and MCL1 amplification detected in ~ 10% of PrCAs appeared mutually exclusive. Finally, MARCH5 and MCL1 modulation made cells more dependent on BCLXL for survival. Overall, whereas the reviewers agreed that while the work is experimentally robust and supports the conclusions made, a number of concerns need to be addressed before the manuscript can be considered further.

Essential revisions:

Please address the following critical points:

1) The novelty of the work was called into question given recent publications in the field showing MCL1 degradation by MARCH5 in a NOXA-dependent manner. Reviewers considered the induction of MARCH5 by EGFR inhibition to be the most novel aspect of the work and the authors are asked to address all of the questions raised by reviewers to expand and enhance this aspect of the work, and also to remove other data not related to how EGFR inhibition modulates NOXA, MCL1, MARCH5 and the ISR.

2) Please address the question of how EGFRi activates the ISR. Is this through PERK, GCN2 or ATF4 by some other mechanism?

3) New experiments should also include key data with additional cell lines as suggested by reviewer #2.

4) Please consider revising the title of the paper and its Abstract to reflect stronger focus on the EGFR inhibitor aspect of the studies. Please refer to comments by reviewer #3 in this regard.

5) Please also consider rewording the manuscript to make it clearer to the reader what is novel, what the key findings are and to remove those aspects found confusing by reviewers.

Without additional experiments addressing more novel aspects of the work, the manuscript will not be favorably reviewed in revisions.

Reviewer #1:

This work by Arai et al., shows that MCL1 levels are reduced by treatment of PrCA cells with EGFR inhibitor erlotinib in a manner dependent on its interaction with NOXA that is transcriptionally induced by erlotinib in an ATF4-dependent manner as part of the integrated stress response, and not via p53. They further show that the degradation of MCL1 in response to erlotinib and NOXA induction is dependent on the MARCH5 E3 Ub ligase, and not on other E3 UBLs previously implicated in MCL1 degradation, such as HUWE1, or by other mitochondrial UBLs such as PARKIN. This effect was specific to treatment of cells with certain kinase inhibitors and not other stresses, and was accompanied by increased levels of MCL1 and other MARCH5 targets. MARCH5 depletion also increased levels of NOXA and BIM that were protected from degradation by interaction with MCL1. Similar mechanisms operated in cells in response to mitochondrial stress inducers like actinonin. Decreased MCL1 induced by erlotinib or actinonin synergized with BH3 mimetics to induce apoptosis. Finally, the authors show that MARCH5 is lost in ~ 5% of primary PrCA and associated with poor prognosis and interestingly that MARCH5 loss and MCL1 amplification detected in ~ 10% of PrCAs may be mutually exclusive and MARCH5 and MCL1 appeared to be co-dependent in CRISPR library screens from the Broad possibly via buffering BAX and making cells more dependent on BCLXL for survival.

The quality of the data presented is strong, well controlled and overall is convincing. The major concern is with the extent to which this work moves the field forward. Novel aspects of the current work depend on showing activation of MARCH5 and NOXA by the ISR but ATF4-mediated induction of NOXA was previously shown. It is noted that a recent publication in Cell Death and Differentiation has shown that MCL1 degradation is mediated by MARCH5 in a NOXA dependent manner in response to anti-mitotics

(https://www.ncbi.nlm.nih.gov/pubmed/32015503) and that MARCH5 regulation of MCL1 has been reported previously in other studies also:

https://www.ncbi.nlm.nih.gov/pubmed/27932492,

https://www.ncbi.nlm.nih.gov/pubmed/26910119

The work also leaves key questions unanswered such as how EGFRi increases the MARCH5 -MCL1 interaction while simultaneously increasing the MCL1NOXA interaction? Is this happening at the mitochondria? The authors would do well to perform cell staining for mitochondria and additional assays to assess how MARCH5 is affecting mitochondrial function, especially since MARCH5 has also been implicated in mitochondrial dynamics and mitophagy which is not addressed here as a means to explaining how MARCH5 may affect tumor cell sensitivity to BH3 mimetics etc. Given that MCL1 itself affects mitochondrial respiration, these analyses are important.

In summary, it is not clear to this reviewer that this work rises to the impact level expected for *eLife* as it currently stands.

Reviewer #2:

This manuscript investigates the mechanistic basis for egfr inhibition dependent MCL1 degradation, finding that EGFR inhibition upregulates the BH3-only protein NOXA dependent on the integrated stress response (ISR), this in turn leads to MCL1 degradation dependent on the mitochondrial E3 ligase March5. Presumably NOXA bound to MCL1 makes it a more amenable to MARCH5 ub ligase activity. The authors then demonstrate that other activators of the ISR, namely mito stresses can affect a similar pathway. Finally, they show that loss of MARCH5 sensitises cells to BCLXL targeting, suggesting potential targeting of this node in MARCH5 deficient cancers.

In my opinion, the data support the authors' conclusions, while some findings have been made elsewhere (as cited by the authors), the majority of this study is novel and timely. I have only a few recommendations:

1) The study has been entirely carried out LNCap cells, raising the question of how generally applicable the conclusions are from the work presented. While I am not suggesting the whole study be repeated across multiple lines, in my view it is important to address the NOXA and MARCH5 dependency of egfr inhibitor MCL1 degradation in additional cell lines.

2) It’s unclear to me why egfr inhibition should engage an ISR, the authors should discuss this.

Reviewer #3;

While in general this manuscript is interesting, it has a lot of data and I wonder whether it would be better to delete some and reduce the complexity. One example is the mitochondrial-targeting agents which seem somewhat tangential and confusing with respect to ISR and NOXA induction which should be ER dependent rather than mitochondrial. The manuscript also contains a lot of speculation on various issues that detracts from its major conclusions (for example, high concentrations of inhibitors). Finally, the role of MARCH5/MCL1/NOXA is not novel, and while the authors do reference the prior paper (Subramanian et al., 2016), they still state in the Discussion "we identified the mitochondria associated ubiquitin ligase MARCH5 as the primary mediator of this NOXA dependent MCL1 degradation" without giving credit to the prior report that concluded the same.

What is novel in this paper is the use of EGFR inhibitors (and other tyrosine kinase inhibitors) to induce NOXA, and that is the focus of this paper. Accordingly, I find the title misleading. A better title might be: EGFR inhibitors mediate MCL degradation that is dependent on the integrated stress response, induction of NOXA and the ubiquitin ligase MARCH5. Furthermore, the Abstract needs to be revised to reflect this. This is important because my original reading of the Abstract led me to believe this was going to use very different inducers of ISR/UPR such as protease inhibitors.

The argument regarding high concentrations of MCL1 inhibitors is confusing, particularly if this is an off-target effect. Much lower concentrations inhibit MCL1 as judged by its stabilization and dissociation of BIM and NOXA. Therefore, if it takes much higher concentrations (and 20 µM is very high for these agents), it seems unlikely to be acting as solely an MCL1 inhibitor. It might make more sense not to include high concentrations which would give different conclusions, and certainly not conclude these might be selective for MCL1 at those concentrations.

How does and EGFRi signal to the ISR? Or perhaps more critically how does EGFRi activate PERK. Some suggestion is warranted.

---

## [Author Response]

Reviewer #1:This work by Arai et al., shows that MCL1 levels are reduced by treatment of PrCA cells with EGFR inhibitor erlotinib in a manner dependent on its interaction with NOXA that is transcriptionally induced by erlotinib in an ATF4-dependent manner as part of the integrated stress response, and not via p53. They further show that the degradation of MCL1 in response to erlotinib and NOXA induction is dependent on the MARCH5 E3 Ub ligase, and not on other E3 UBLs previously implicated in MCL1 degradation, such as HUWE1, or by other mitochondrial UBLs such as PARKIN. This effect was specific to treatment of cells with certain kinase inhibitors and not other stresses, and was accompanied by increased levels of MCL1 and other MARCH5 targets. MARCH5 depletion also increased levels of NOXA and BIM that were protected from degradation by interaction with MCL1. Similar mechanisms operated in cells in response to mitochondrial stress inducers like actinonin. Decreased MCL1 induced by erlotinib or actinonin synergized with BH3 mimetics to induce apoptosis. Finally, the authors show that MARCH5 is lost in ~ 5% of primary PrCA and associated with poor prognosis and interestingly that MARCH5 loss and MCL1 amplification detected in ~ 10% of PrCAs may be mutually exclusive and MARCH5 and MCL1 appeared to be co-dependent in CRISPR library screens from the Broad possibly via buffering BAX and making cells more dependent on BCLXL for survival.The quality of the data presented is strong, well controlled and overall is convincing. The major concern is with the extent to which this work moves the field forward. Novel aspects of the current work depend on showing activation of MARCH5 and NOXA by the ISR but ATF4-mediated induction of NOXA was previously shown.

We appreciate the reviewer’s comment that the quality of the data is strong, and agree that some of our findings in linking the chain of events from kinase inhibitors to the ISR and to MCL1 degradation are not entirely novel. However, while the ATF4 induction of NOXA has been described previously (which we reference), the acute effects of ISR and ATF4 are generally to increase chaperone and related proteins to resolve stress, with NOXA induction being described as a late event after prolonged stress. In contrast, we found that the induction of NOXA was acute and dramatic, which we believe is a novel finding and indicates that activation of the ISR in some contexts may acutely activate apoptotic versus cell survival pathways. We speculate that this may reflect effects on pathways that intersect with the ISR and modulate the function of ATF4, which is an area for further studies.

It is noted that a recent publication in Cell Death and Differentiation has shown that MCL1 degradation is mediated by MARCH5 in a NOXA dependent manner in response to anti-mitotics(https://www.ncbi.nlm.nih.gov/pubmed/32015503) and that MARCH5 regulation of MCL1 has been reported previously in other studies also:https://www.ncbi.nlm.nih.gov/pubmed/27932492,https://www.ncbi.nlm.nih.gov/pubmed/26910119

We cited in the manuscript the latter two references noted above. The first was focused on mitochondrial dynamics and noted that MCL1 was increased in response to MARCH5 depletion (Reference 27932492, Novel Regulatory Roles of Mff and Drp1 in E3 Ubiquitin Ligase MARCH5-dependent Degradation of MiD49 and Mcl1 and Control of Mitochondrial Dynamics). The second study found that MARCH5 depletion could increase both NOXA and MCL1 and sensitize to ABT-737, targeting BCLXL (Reference 26910119, Inhibition of MARCH5 Ubiquitin Ligase Abrogates MCL1-dependent Resistance to BH3 Mimetics via NOXA). It should also be noted that previous studies that we cited have shown that NOXA can bind to MCL1 and enhance MCL1 degradation, and suggest that this is by some mechanism coupled to the degradation of NOXA. Our results are certainly consistent with these previous findings, but are distinct as they establish MARCH5 as the primary ubiquitin ligase regulating MCL1 levels in response to cellular stress. This is significant as multiple ubiquitin ligases have been reported to mediate the degradation of MCL1, but it appears that they each may function to regulate MCL1 in specific contexts or in response to particular perturbations. Related to this theme, the first reference noted above by the reviewer (Reference 32015503, MARCH5-dependent Degradation of MCL1/NOXA Complexes Defines Susceptibility to Antimitotic Drug Treatment) indicates that MARCH5 is a major mediator of MCL1 degradation during mitotic arrest. This study came out subsequent to our submission, and we now cite it in the revised manuscript as a second context in which MARCH5 functions as a major regulator of MCL1.

The work also leaves key questions unanswered such as how EGFRi increases the MARCH5 -MCL1 interaction while simultaneously increasing the MCL1NOXA interaction? Is this happening at the mitochondria? The authors would do well to perform cell staining for mitochondria and additional assays to assess how MARCH5 is affecting mitochondrial function, especially since MARCH5 has also been implicated in mitochondrial dynamics and mitophagy which is not addressed here as a means to explaining how MARCH5 may affect tumor cell sensitivity to BH3 mimetics etc. Given that MCL1 itself affects mitochondrial respiration, these analyses are important.

We are in complete agreement with the reviewer that mitochondrial dynamics may modulate the interaction between MARCH5 and MCL1. To address this issue we carried out a series of studies (shown in Figure 4) to address whether erlotinib altered MARCH5 expression, phosphorylation, or activity (based on levels of the MARCH5 substrate MiD49), whether it altered mitochondrial association of MARCH5 or MCL1, and whether it altered mitochondrial function. As shown in Figure 4, we could not detect any alterations in MARCH5, but did find that erlotinib (and lapatinib) caused an increase in basal oxygen consumption (Figure 4G, I). Therefore, we carried out parallel studies to determine whether directly perturbing mitochondrial function would drive MARC5 dependent degradation of MCL1. As shown in Figure 5, we found that three different mitochondrial targeted drugs rapidly decreased MCL1. However, similarly to the effect of erlotinib, this was mediated by ISR activation and increased NOXA. Based on these findings, we conclude that MARCH5 is not being altered by erlotinib or ISR activation, and that its increased degradation of MCL1 is being driven by increased NOXA binding to MCL1. Importantly, NOXA binding to MCL1 appears to stabilize a conformational of MCL1 that has been reported to enhance binding to HUWE1, and we hypothesize that this conformational change is the basis for the increased interaction with MARCH5 (which our study indicates to be the major ubiquitin ligase recognizing this conformation).

While we believe that the above explanation partially addressed the reviewer’s comments (explains the basis for increased MCL1 binding to MARCH5), we agree that there is a broader question of how MARCH5 affects tumor cell sensitivity to BH3 mimetics, and to apoptosis in general. MARCH5 regulation of MCL1 is certainly one factor, but given the established role of MARCH5 in mitochondrial dynamics and mitophagy, it is likely that MARCH5 has additional roles in apoptosis. Indeed, it is highly significant that MARCH5 depletion, although causing an increase in MCL1, sensitizes tumor cells to BCLXL inhibition. We suggest in that this reflects a distinct function of MARCH5 (or possibly MARCH5 associated with MCL1) in preventing mitochondrial accumulation of BAX. We are pursuing this area, which we believe will yield additional novel and significant insights, but this is beyond the scope of the current study.

Reviewer #2:This manuscript investigates the mechanistic basis for egfr inhibition dependent MCL1 degradation, finding that EGFR inhibition upregulates the BH3-only protein NOXA dependent on the integrated stress response (ISR), this in turn leads to MCL1 degradation dependent on the mitochondrial E3 ligase March5. Presumably NOXA bound to MCL1 makes it a more amenable to MARCH5 ub ligase activity. The authors then demonstrate that other activators of the ISR, namely mito stresses can affect a similar pathway. Finally, they show that loss of MARCH5 sensitises cells to BCLXL targeting, suggesting potential targeting of this node in MARCH5 deficient cancers.In my opinion, the data support the authors' conclusions, while some findings have been made elsewhere (as cited by the authors), the majority of this study is novel and timely. I have only a few recommendations:1) The study has been entirely carried out LNCap cells, raising the question of how generally applicable the conclusions are from the work presented. While I am not suggesting the whole study be repeated across multiple lines, in my view it is important to address the NOXA and MARCH5 dependency of egfr inhibitor MCL1 degradation in additional cell lines.

We agree that it is important to confirm that any results can be generalized. We had shown in the supplemental data that MARCH5 depletion increases MCL1 in multiple cell lines (Figure 3—figure supplement 1G-L). We also showed the MARCH5 dependency for MCL1 degradation in response to erlotinib in PC3 cells as well as in LNCaP cells. In response to the reviewer’s comments, we are adding data that similarly shows MARCH5 dependency for MCL1 degradation in a third prostate cancer cell line (DU145 cells) (Figure 3—figure supplement 3I).

2) It’s unclear to me why egfr inhibition should engage an ISR, the authors should discuss this.

We did indicate in the Discussion that erlotinib, while being highly specific for EGFR and not targeting other known kinases, may nonetheless interact weakly with multiple proteins containing ATP binding sites, including many mitochondrial proteins involved in ATP synthesis. Of note, while neither erlotinib nor lapatinib decreased maximal oxygen consumption, indicating that mitochondria function was not impaired, they did increase basal oxygen consumption, indicating a shift from fermentation to oxidative phosphorylation. This could in part be a downstream effect of EGFR inhibition, but may also be due to drug effects on other ATP binding proteins involved in metabolism. In any case, we would hypothesize that the kinase inhibitors, similarly to the mitochondria targeted agents we examined, are causing a metabolic stress that is engaging the ISR. In the revised manuscript we have expanded our discussion of this issue.

Reviewer #3;While in general this manuscript is interesting, it has a lot of data and I wonder whether it would be better to delete some and reduce the complexity. One example is the mitochondrial-targeting agents which seem somewhat tangential and confusing with respect to ISR and NOXA induction which should be ER dependent rather than mitochondrial.

As MARCH5 is a mitochondrial associated ubiquitin ligase with a role in regulating mitochondrial dynamics and mitophagy (as noted by reviewer 1), we believed it was important to address whether erlotinib was affecting mitochondrial dynamics and subsequently MARCH5 activity. Our approach was to directly assess for effects on MARCH5 in Figure 4 (expression, phosphorylation, and activity), and to determine whether perturbing mitochondrial dynamics with mitochondrial-targeting agents would similarly enhance MCL1 degradation (Figure 5). We found that these latter agents did indeed drive MARCH5 dependent MCL1 degradation, but it was due to induction of an ISR rather than through a direct effect on MARCH5. Of note, while ISR activation through PERK reflects ER stress, the ISR can be activated by other mechanisms and there are increasing data linking mitochondrial stress to the ISR. Based on these considerations, we believe that the mitochondrial-targeting agent data do provide significant insight and should be retained.

The manuscript also contains a lot of speculation on various issues that detracts from its major conclusions (for example, high concentrations of inhibitors). Finally, the role of MARCH5/MCL1/NOXA is not novel, and while the authors do reference the prior paper (Subramanian et al., 2016), they still state in the Discussion "we identified the mitochondria associated ubiquitin ligase MARCH5 as the primary mediator of this NOXA dependent MCL1 degradation" without giving credit to the prior report that concluded the same.

The study in Subramanian et al., 2016 (Inhibition of MARCH5 Ubiquitin Ligase Abrogates MCL1-dependent Resistance to BH3 Mimetics via NOXA) found that MARCH5 depletion could increase both NOXA and MCL1 and sensitize to ABT-737, targeting BCLXL. As noted by the reviewer, we did cite and discuss this study in the Results and Discussion sections, as its findings were significant and clearly related to our study. However, we did not cite it at the place indicated above by the reviewer as the study concluded that NOXA was stabilizing MCL1 rather than driving its degradation, as indicated below in the Results section of their study:

“However, NOXA loss also robustly attenuated the induction of MCL1 that we observed upon MARCH5 knockdown (Figure 4B). Together, these data indicate that NOXA is required for maximal stabilization of MCL1 following loss of MARCH5. These data are consistent with other reports of NOXA-dependent stabilization of MCL1.”

What is novel in this paper is the use of EGFR inhibitors (and other tyrosine kinase inhibitors) to induce NOXA, and that is the focus of this paper. Accordingly, I find the title misleading. A better title might be: EGFR inhibitors mediate MCL degradation that is dependent on the integrated stress response, induction of NOXA and the ubiquitin ligase MARCH5. Furthermore, the Abstract needs to be revised to reflect this. This is important because my original reading of the Abstract led me to believe this was going to use very different inducers of ISR/UPR such as protease inhibitors.

We agree that the rapid and robust induction of NOXA by EGFR and other kinase inhibitors, as well as by mitochondrial-targeted agent is novel. However (as noted above and in the response to reviewer 1), while MARCH5 has been implicated previously as a ubiquitin ligase for MCL1, we also believe that its identification as the major mediator of stress-induced and NOXA-dependent MCL1 degradation is novel, and that this is the most important aspect of the study. Therefore, while the title suggested by the reviewer would encompass our findings, we would be concerned that it would suggest that the major focus is on EGFR inhibitors. However, in response to this suggestion we have modified the title to reflect kinase inhibitor induction of NOXA. If the editors and reviewers are agreeable, the new title in this revision is “MARCH5 Mediates NOXA-Dependent MCL1 Degradation Driven by Kinase Inhibitors and Integrated Stress Response Activation”. We have also modified the Abstract to indicate that NOXA is being induced by the ISR in response to kinase inhibitors.

The argument regarding high concentrations of MCL1 inhibitors is confusing, particularly if this is an off-target effect. Much lower concentrations inhibit MCL1 as judged by its stabilization and dissociation of BIM and NOXA. Therefore, if it takes much higher concentrations (and 20 µM is very high for these agents), it seems unlikely to be acting as solely an MCL1 inhibitor. It might make more sense not to include high concentrations which would give different conclusions, and certainly not conclude these might be selective for MCL1 at those concentrations.

We would be reluctant to exclude the 20 mM data points as it is possible that these high concentrations may be required to abrogate MCL1 binding to BAK due the high levels of MCL1, and readers may therefore want to know what happens at these high concentrations. However, we agree with the reviewer that this could be off-target, and this is noted in the text. To clarify this issue, we have revised this section in the Results to more clearly distinguish between a possible on-target mechanism versus the perhaps more likely interpretation that these cells do not have an increased dependence on MCL1, despite its high level expression.

How does and EGFRi signal to the ISR? Or perhaps more critically how does EGFRi activate PERK. Some suggestion is warranted.

We did indicate in the Discussion that erlotinib, while being highly specific for EGFR and not targeting other known kinases, may nonetheless interact weakly with multiple proteins containing ATP binding sites, including many mitochondrial proteins involved in ATP synthesis. Of note, while neither erlotinib nor lapatinib decreased maximal oxygen consumption, indicating that mitochondria function was not impaired, they did increase basal oxygen consumption, indicating a shift from fermentation to oxidative phosphorylation. This could in part be a downstream effect of EGFR inhibition, but may also be due to drug effects on other ATP binding proteins involved in metabolism. In any case, we would hypothesize that the kinase inhibitors, similarly to the mitochondria targeted agents we examined, are causing a metabolic stress that is engaging the ISR. In the revised manuscript we have expanded our discussion of this issue.

With respect to PERK, we describe in the text that the ISR can be activated by several mechanisms including PERK (a sensor for ER stress) and GCN2, which responds to metabolic stresses including amino acid starvation. Alterations in ATP binding and metabolic shifts as above could in principal have effects on ATP dependent processes in the ER or cytosol. To further address the mechanism, and in particular whether the ISR is being activated through PERK or GCN2, we used siRNA targeting PERK or GCN2, or both in combination, and determined whether these treatments impaired the downregulation of MCL1 in response to erlotinib. As shown in the new Figure 2—figure supplement 1, siRNA targeting PERK or GCN2 did not prevent the erlotinib-mediated decrease in MCL1, but the combined siRNA did prevent the decrease in MCL1. Based on these data, in the revised text we now suggest that erlotinib is impacting ATP dependent functions in both the ER and cytosol.